# Architecture of the chikungunya virus replication organelle

**Timothée Laurent**[1,2,3,4], **Pravin Kumar**[1,2,3,4], **Susanne Liese**[5,6], **Farnaz Zare**[1,2,3,4], **Mattias Jonasson**[1,2,3,4], **Andreas Carlson**[6]*, **Lars-Anders Carlson**[1,2,3,4]*

[1]Department of Medical Biochemistry and Biophysics, Umeå University, Umeå, Sweden; [2]Molecular Infection Medicine Sweden, Umeå University, Umeå, Sweden; [3]Wallenberg Centre for Molecular Medicine, Umeå University, Umeå, Sweden; [4]Umeå Centre for Microbial Research (UCMR), Umeå, Sweden; [5]Max Planck Institute for the Physics of Complex Systems, Dresden, Germany; [6]Department of Mathematics, Mechanics Division, University of Oslo, Oslo, Norway

**Abstract** *Alphaviruses* are mosquito-borne viruses that cause serious disease in humans and other mammals. Along with its mosquito vector, the *Alphavirus* chikungunya virus (CHIKV) has spread explosively in the last 20 years, and there is no approved treatment for chikungunya fever. On the plasma membrane of the infected cell, CHIKV generates dedicated organelles for viral RNA replication, so-called spherules. Whereas structures exist for several viral proteins that make up the spherule, the architecture of the full organelle is unknown. Here, we use cryo-electron tomography to image CHIKV spherules in their cellular context. This reveals that the viral protein nsP1 serves as a base for the assembly of a larger protein complex at the neck of the membrane bud. Biochemical assays show that the viral helicase-protease nsP2, while having no membrane affinity on its own, is recruited to membranes by nsP1. The tomograms further reveal that full-sized spherules contain a single copy of the viral genome in double-stranded form. Finally, we present a mathematical model that explains the membrane remodeling of the spherule in terms of the pressure exerted on the membrane by the polymerizing RNA, which provides a good agreement with the experimental data. The energy released by RNA polymerization is found to be sufficient to remodel the membrane to the characteristic spherule shape.

*For correspondence:
acarlson@math.uio.no (AC);
lars-anders.carlson@umu.se (L-AC)

**Competing interest:** The authors declare that no competing interests exist.

## Editor's evaluation

Chikungunya virus is a very important human pathogen, and research on the architecture of its replication/transcription organelle holds great promise for the development of future therapies. Laurent and colleagues advanced this field by providing pioneering low-resolution 3D structures of the membrane-bound viral protein complex and the viral RNA content of this organelle in situ. In addition, they also assessed the lipid requirements for membrane interaction of the primary viral membrane anchor of this complex, nsP1, in vitro.

## Introduction

Chikungunya is a mosquito-borne disease characterized by a rapid onset of fever, followed by debilitating joint pains and arthritis that can last for months or years (*Weaver and Lecuit, 2015*; *Burt et al., 2017*). It is severely underdiagnosed, but suspected cases have surpassed 500,000/year in several recent years (https://www.who.int/news-room/fact-sheets/detail/chikungunya). The causative agent of chikungunya is chikungunya virus (CHIKV), a positive-sense single-stranded RNA virus of the *Alphavirus* genus (family *Togaviridae*). In the last two decades, CHIKV has spread rapidly, far beyond

its probable origins in east Africa, to cause large outbreaks in Asia and the Americas. One reason for this is its adaptation to a new mosquito host, *Aedes albopictus*, which inhabits more temperate regions (*Tsetsarkin et al., 2007*; *Vazeille et al., 2007*). In addition to CHIKV, a plethora of pathogenic *Alphaviruses* exist, and their utilization of different mosquito species highlights the potential for new variants to arise and spread. There are no approved vaccines or antivirals against any *Alphavirus*-caused diseases.

The replication of the *Alphavirus* genome takes place in a virus-induced RNA replication organelle, also known as a 'spherule' or 'replication complex.' This organelle is formed as an outward-facing plasma membrane bud with a diameter of 50–80 nm (*Ahola et al., 2021*). The size of the membrane bud has been shown to depend on the length of the viral genome (*Kallio et al., 2013*). The bud is thought to have a stable, open neck that connects it to the cytoplasm, and this high-curvature membrane shape persists for several hours in the infected cell during active RNA production. The viral nsPs are thus thought to serve the additional role of maintaining this peculiar membrane shape while replicating the viral RNA.

The *Alphavirus* genome codes for four non-structural proteins (nsP1–nsP4), initially produced as one polyprotein, with distinct functions in the viral genome replication (*Ahola et al., 2021*; *Rupp et al., 2015*). NsP1 caps the 5' end of the new viral RNA independently of the host-cell capping machinery (*Ahola et al., 2021*). It is the only nsP reported to bind membranes, and its membrane affinity is enhanced by, but not dependent on, a palmitoylation site (*Ahola et al., 2000*). NsP2 has RNA helicase and RNA triphosphatase activity in its N-terminal domain, and its C-terminus harbors a cysteine protease domain which cleaves the polyprotein into individual nsPs. NsP3 has ADP-ribosyl hydrolase activity and interacts with several host-cell proteins (*Götte et al., 2018*). NsP4 is the RNA-dependent RNA polymerase directly responsible for the production of new viral RNA.

Structures have been determined for individual domains of the nsPs (*Law et al., 2019*; *Shin et al., 2012*; *Tan et al., 2021*). Although informative for the function of the individual proteins, the structures generally provide no clues as to how the nsPs spatially coordinate the different steps of the RNA production and the membrane remodeling. One exception is the structure of the isolated, nsP1 protein (*Jones et al., 2021*; *Zhang et al., 2021*). When overexpressed in eukaryotic systems and gently extracted from the plasma membrane, nsP1 was shown to form a ring-shaped dodecamer, displaying its active sites to the inside of the ring and the membrane-binding surfaces to the outside. It was thus suggested that the nsP1 dodecamer may bind at and stabilize the high-curvature membrane neck. This model remains to be tested experimentally, and it is not known how localization of nsP1 at the neck would relate to other protein components in the spherule, the RNA, or the membrane shape.

Here, we use cellular cryo-electron tomography, in vitro reconstitution, and mathematical modeling to provide the first integrated model of the CHIKV spherule. Our findings reveal that nsP1 anchors a large protein complex at the membrane neck and directly recruits nsP2 to the membrane. The lumen of full-sized spherules contains a single copy of the viral genome, and we present a theoretical model that explains how RNA polymerization leads to a membrane remodeling consistent with the shapes observed in the tomograms.

## Results

### Cryo-electron tomography allows visualization of CHIKV spherules at the plasma membrane

We wished to study the structure of the CHIKV spherule in situ in unperturbed cells. The high biosafety level necessitated by CHIKV is typically dealt with by chemical fixation of infected cells prior to electron microscopy. Since this may compromise macromolecular organization, we instead opted to use viral replicon particles (VRPs), which transduce cells with a replication-competent, but capsid protein-deleted, genome that results in a self-limiting single-cycle infection (*Gläsker et al., 2013*). The VRPs express an eGFP (enhanced green fluorescent protein) reporter gene in place of the capsid proteins, which allowed confirmation that a vast majority of the cells grown on EM (electron microscopy) sample grids were transduced and had active viral RNA replication (*Figure 1—figure supplement 1*). Cryo-electron tomograms of the peripheral plasma membrane occasionally showed CHIKV spherules appearing as clusters of balloon-shaped organelles sitting at the plasma membrane (*Figure 1A–B*; *Figure 1—videos 1; 2*). They had a diameter ranging from 50 to 70 nm, consistent with what has

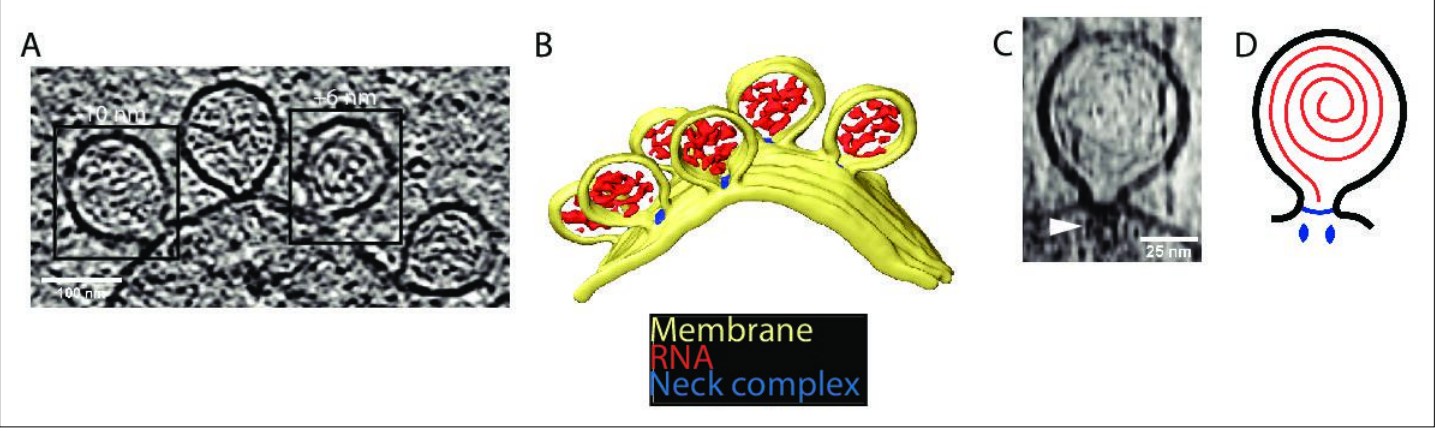

**Figure 1.** Cryo-electron tomography visualizes chikungunya virus (CHIKV) spherules at the plasma membrane. (**A**) Computational slice through a cryo-electron tomogram of a CHIKV viral replicon particle (VRP)-transduced baby hamster kidney (BHK) cell. The two framed insets are offset in the tomogram volume by 7 nm. Scale bar, 100 nm. (**B**) 3D segmentation of the tomogram shown in (**A**). Yellow: plasma membrane, red: viral RNA, and blue: protein complex sitting at the spherule necks. (**C**) Subtomogram containing one spherule. The arrow indicates the densities present at the membrane neck. Scale bar, 25 nm. (**D**) Schematic of an initial model of the organization of a spherule. Black: plasma membrane, red: viral RNA, and blue: protein complex sitting at the spherule necks.

The online version of this article includes the following video and figure supplement(s) for figure 1:

**Figure supplement 1.** Workflow for cryo-electron tomography of chikungunya virus (CHIKV) viral replicon particle (VRP)-transduced cells.

**Figure 1—video 1.** Cryo-electron tomogram of a cluster of chikungunya virus (CHIKV) spherules at the plasma membrane.

https://elifesciences.org/articles/83042/figures#fig1video1

**Figure 1—video 2.** Uncropped cryo-electron tomogram of a cluster of chikungunya virus (CHIKV) spherules at the plasma membrane.

https://elifesciences.org/articles/83042/figures#fig1video2

**Figure 1—video 3.** Tracing of dsRNA filaments in spherules.

https://elifesciences.org/articles/83042/figures#fig1video3

been reported from resin-section EM (*Kujala et al., 2001*). In addition to the membrane topology, the cryo-electron tomograms also revealed filamentous densities coiled on the inside of the membrane buds (*Figure 1A–B*; *Figure 1—video 1*). The position and width of the filaments make it likely that they are viral RNA, possibly in its dsRNA replicative intermediate. We next turned to the stabilization of membrane curvature. In principle, the high-curvature membrane of the CHIKV spherule could be stabilized either by protein binding throughout the curved membrane or by specific stabilization of the membrane neck. From visual inspection, there was no consistent pattern of protein coating over the curved surface of the membrane bud. On the other hand, in all imaged spherules, we observed a macromolecular complex sitting at the membrane neck (*Figure 1A–B*). In well-resolved individual spherules, the complex seemed to be bipartite with a base pinching the neck of the spherule and a second part protruding toward the cytoplasm of the cell (*Figure 1C*). Taken together, these data suggest that the CHIKV spherule consists of a membrane bud filled with viral RNA and has a macromolecular complex gating the opening of this bud to the cytoplasm (*Figure 1D*).

## Subtomogram averaging determines the position of nsP1 in a larger neck complex

We were interested in investigating the structure of the protein complex sitting at the membrane neck. A 34 Å subtomogram average was calculated (*Figure 2—figure supplement 1*) from 64 spherules without imposing any symmetry. It revealed that the complex is composed of two parts: a membrane-bound 'base' and a barrel-like 'crown' (*Figure 2A–D*). The base fits the membrane neck snugly (*Figure 2A–B*). The crown is composed of three rings and protrudes toward the cytoplasm (*Figure 2A–C*). At the current resolution, there is no visible connection between the base and the crown. A third component of the neck complex is a central density protruding from the base, through the crown toward the cytoplasm. It appears more diffuse than the base and the crown.

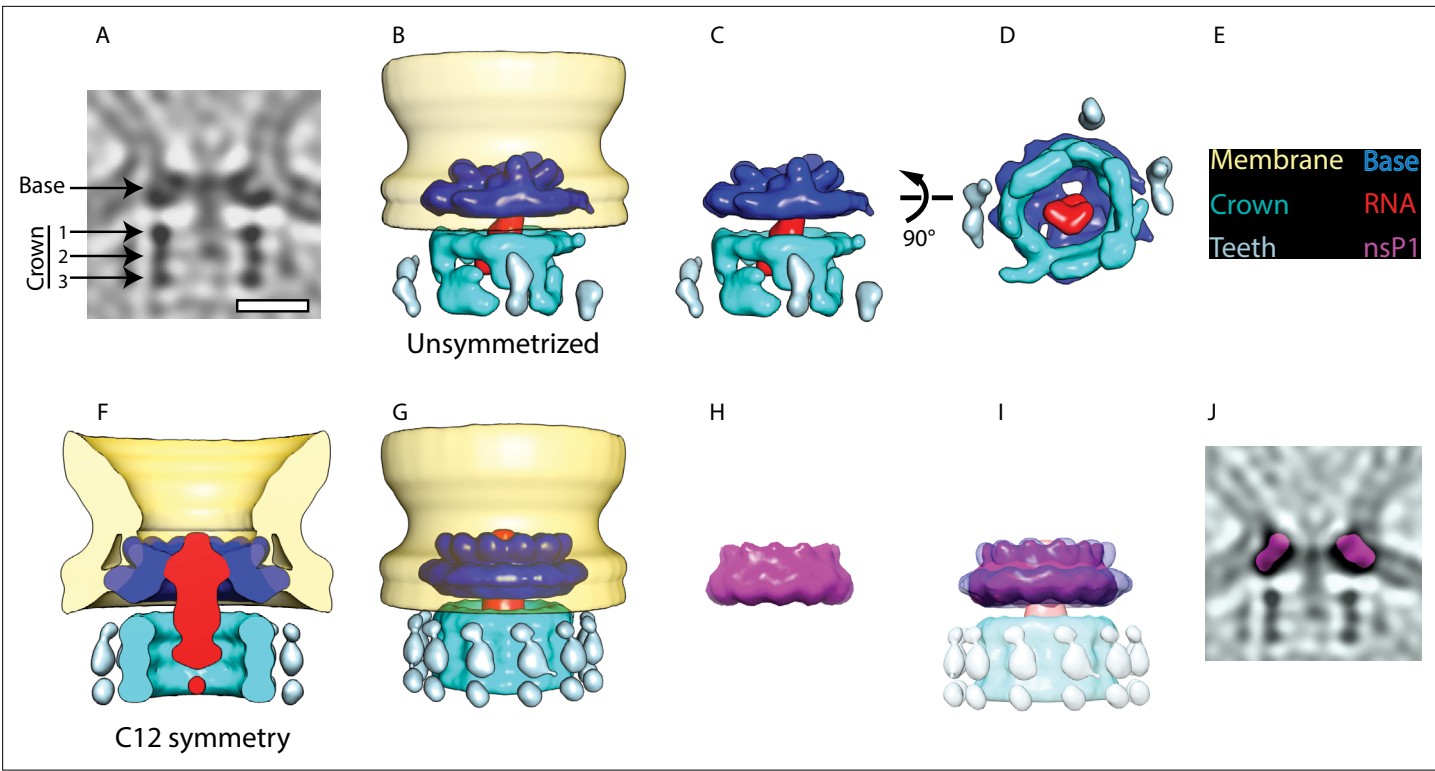

**Figure 2.** Subtomogram averaging reveals the multipartite nature of the neck complex. (**A**) Central slice through the unsymmetrized subtomogram average of the neck complex, low-pass filtered to its Gold-standard resolution 34 Å. Arrows indicate densities that are referred to as the 'base' and the 'crown.' The crown is located on the cytoplasmic side and is composed of three rings. Density is black. Scale bar, 100 Å. (**B**) 3D segmentation of the unsymmetrized subtomogram average shown in (**A**). The spherule membrane (yellow) is radially symmetrized for clarity. Dark blue: base, red: putative RNA, cyan: crown, and light blue: teeth. (**C and D**) Two views of the neck complex related by the indicated rotation. (**E**) Color key for all panels. (**F**) Cross-section through the subtomogram average of the neck complex with C12 symmetry imposed, low-pass filtered to its Gold-standard resolution 28 Å. (**G**) Surface view corresponding to (**F**). (**H**) The structure of the isolated nsP1 (from *Jones et al., 2021*) low-pass filtered to the resolution of our average. (**I**) Superimposition of nsP1 onto the base of the protein neck complex. (**J**) Slice through the unsymmetrized subtomogram average, as in (**A**), with a slice of the fitted nsP1 superimposed on the base of the complex.

The online version of this article includes the following figure supplement(s) for figure 2:

**Figure supplement 1.** Subtomogram averaging of the neck complex.

We hypothesized that the base of the neck complex may be nsP1, the only nsP with known membrane-binding motifs. The recent structures of nsP1 revealed a ring-shaped dodecamer with a similar dimension to the base of the neck complex (*Jones et al., 2021*; *Zhang et al., 2021*). For comparison, we imposed 12-fold symmetry on our neck complex (*Figure 2F–G*) and low-pass filtered the published nsP1 structure to the 28 Å resolution of the 12-fold symmetrized average (*Figure 2H*). An overlay of these two showed a close match in size and shape of the isolated nsP1 and the base of the neck complex (*Figure 2I*). The best fit of nsP1 into the neck complex is such that the narrow side of the nsP1 ring, carrying the membrane-association sites, is in direct contact with the membrane. We further verified that nsP1 fits the unsymmetrized neck complex average (*Figure 2J*). This overlay indicated that there may be additional densities bound to the inside of the nsP1 ring in the full-neck complex as compared to the heterologously expressed nsP1.

There was not sufficient signal in the subtomogram average to experimentally determine the rotational symmetry in the crown part of the neck complex. But the main features were consistent between the unsymmetrized and the C12 averages: the crown consists of three stacked rings of equal diameter (*Figure 2A and F*), and there is weaker but consistent density for peripheral structures ('teeth') surrounding the rings (*Figure 2B, C and G*). At the current resolution, the components forming the crown and teeth cannot be identified from the subtomogram average. However, based on their volume of 1500–1700 nm³, we estimate them to have a molecular mass of 1.2–1.4 MDa. At the center of the neck complex, extending out from nsP1 through the crown is a rod-like density that is the

only candidate to be the new viral RNA leaving the spherule. In summary, the subtomogram average suggests that nsP1 forms the base of a larger neck complex that extends toward the cytoplasm with a barrel-like structure that may funnel new viral RNA out from the spherule lumen.

## NsP1 recruits nsP2 to membranes containing monovalent anionic lipids

The subtomogram average suggested that nsP1 acts as the assembly platform for other viral nsPs. To test this experimentally, we took an in vitro reconstitution approach. We purified recombinant CHIKV nsP1 to homogeneity (*Figure 3—figure supplement 1*). To test whether a monomeric nsP1 can bind the membrane prior to oligomerization, we used the monomeric fraction of nsP1 and synthetic liposomes in a multilamellar vesicle (MLV) pulldown assay (*Figure 3—figure supplement 1*). In the absence of any negatively charged lipids, nsP1 did not bind appreciably to the vesicles (*Figure 3A*). Semliki forest virus (SFV) nsP1 has been reported to associate with phosphatidyl serine (PS), an abundant lipid on the inner leaflet of the plasma membrane (*Ahola et al., 1999*). Thus, we next decided to include PS in the MLVs. This revealed that nsP1 has concentration-dependent binding to PS-containing membranes (*Figure 3A*, *Figure 3—figure supplement 2*). The pulldowns were repeated in the presence of other monovalent anionic glycerophospholipids (phosphatidyl glycerol [PG] and phosphatidylinositol [PI]) to test whether the binding was specific to PS or more generally dependent on membrane charge. NsP1 showed very similar, concentration-dependent binding to PG- and PI-containing membranes (*Figure 3A*). We then studied the interaction of nsP1 with phosphoinositides (PIPs), lipids that serve as membrane identity markers and may thus be involved in targeting the spherule assembly to a specific membrane. We compared two PIPs: the predominantly Golgi-resident PI (4)P and the predominantly plasma membrane-resident PI(4,5) $P_2$. Curiously, nsP1 had a higher affinity to membranes containing low PIP concentrations and almost no affinity for membranes with higher concentration of these lipids (*Figure 3B–C*). For each PIP, we observed weaker membrane association than to membranes containing monovalent anionic lipids (*Figure 3A–C*). As an alternative approach, we visualized the interaction of nsP1 with giant unilamellar vesicles (GUVs) using confocal microscopy. No accumulation of fluorescent nsP1 was seen on GUVs consisting of phosphatidyl choline and cholesterol (a net-uncharged membrane). On the other hand, the equivalent charge density introduced in the form of either 20% PS or 5% PI(4,5)$P_2$ led to visible binding of nsP1 to the surface of GUVs (*Figure 3D*). 20% of the PI(4,5)$P_2$-containing GUVs were positive for nsP1 binding, whereas 50% of PS-containing GUVs were positive, paralleling the MLV pulldown results (*Figure 3E*).

The MLV pulldown assay was then extended to investigate if nsP1 can anchor other nsPs to the membrane. Both nsP3 and nsP4 have long disordered regions which make it challenging to obtain high-quality monodisperse protein. However, we were able to purify recombinant full-length nsP2 to homogeneity and obtained a monomeric protein (*Figure 3—figure supplement 1*). In the pulldown assay, nsP2 had no affinity to membranes containing 70% PS. However, nsP2 was recruited to the membrane by nsP1 in a concentration-dependent manner (*Figure 3F–G*). Taken together, these data show that the recruitment of nsP1 to membranes dependent mainly on monovalent anionic lipids and that nsP1 can serve as a docking place for nsP2, which has no membrane affinity of its own (*Figure 3H*).

## Full-size spherules contain a single copy of the genome that is largely present in dsRNA form

Turning next to the RNA component of the spherule, we reasoned that the visible filaments in the spherule lumen would allow an estimation of the total copy number of viral RNA within single spherules. The filaments were frequently observed to be relatively straight over a large fraction of the spherule lumen, which is more compatible with the persistence length of dsRNA (63 nm) than that of single-stranded RNA (1 nm) (*Figure 1C*; *Figure 1—video 1*; *Abels et al., 2005*; *Hyeon et al., 2006*). A dsRNA form of the filaments would also be consistent with immunofluorescence observations of high loads of dsRNA at *Alphavirus* replication sites (*Spuul et al., 2011*) and the positioning of the helicase nsP2 at the neck complex where it could serve a role in unwinding dsRNA (*Figure 3F–H*; *Das et al., 2014*). Using an automated filament tracing algorithm, developed to trace cytoskeletal and other filaments in cryo-electron tomograms (*Rigort et al., 2012*), we were able to trace long continuous stretches of dsRNA in the spherule lumen (*Figure 4A*, *Figure 4—figure supplement 1*). The traced model agreed well with filamentous densities seen in the tomograms, and the total filament length was robust over a wide range of parameter values (*Figure 1—video 3*, *Figure 4—figure*

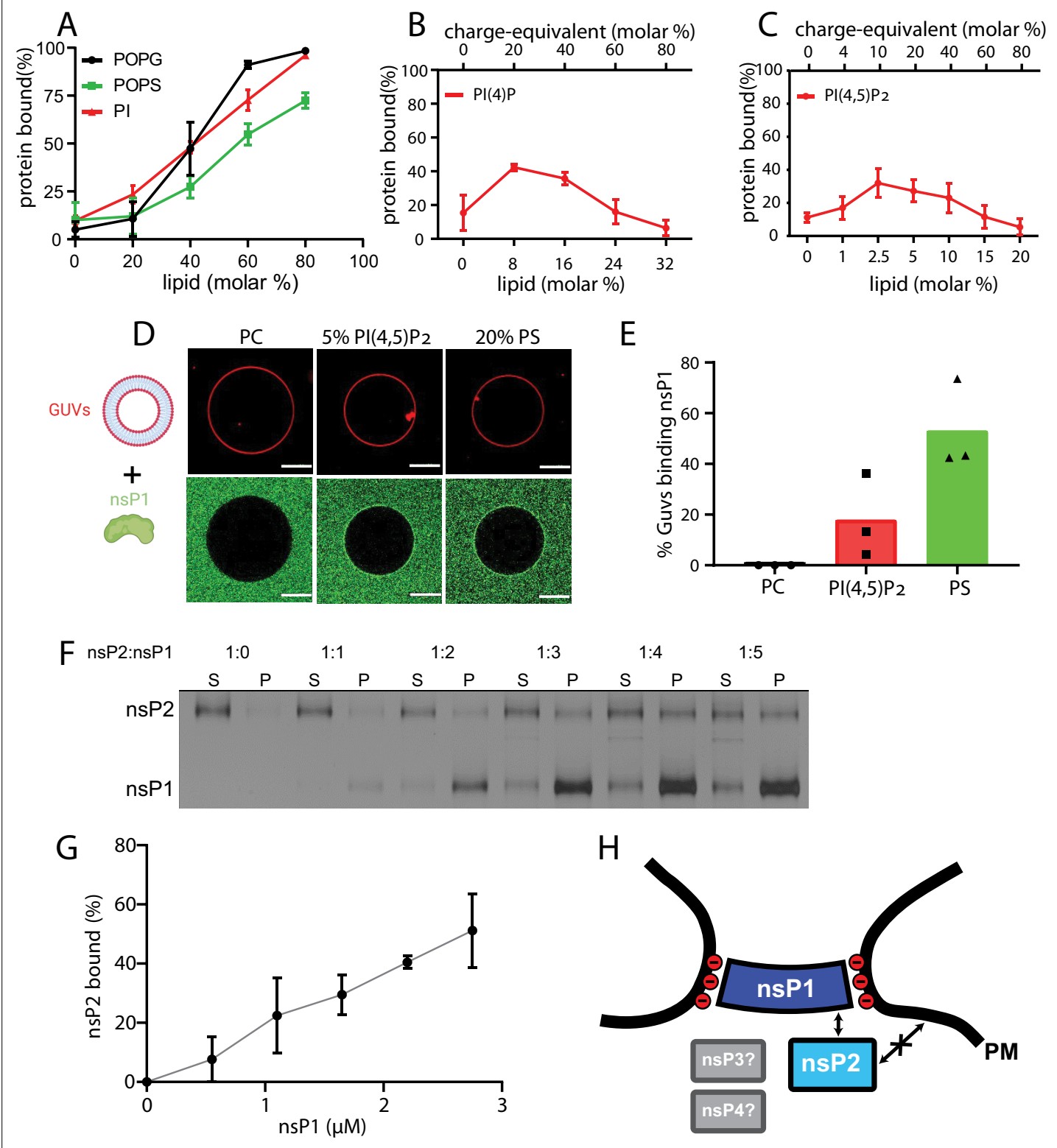

**Figure 3.** NsP1 binds to membranes containing monovalent anionic lipids and recruits nsP2 in a concentration-dependent manner. (**A–C**) Copelletation of nsP1 with multilamellar vesicles (MLVs) with varying percentages of the anionic phospholipids 1-palmitoyl-2-oleoyl-sn-glycero-3-phospho-l-serine (POPS), 1-palmitoyl-2-oleoyl-sn-glycero-3-phospho-(1'-rac-glycerol) (POPG), phosphatidylinositol (PI) (**A**), PI(4)P (**B**), and PI(4,5)P₂ (**C**) in a background of 1-palmitoyl-2-oleoyl-sn-glycero-3-phosphocholine (POPC) and 20% cholesterol. Representative example gels shown in *Figure 3—figure supplement 2*. The percentage of protein associated with membranes was quantitated from gels and plotted. Each plot represents the mean ± SD of three

*Figure 3 continued on next page*

*Figure 3 continued*

independent replicates. (**D**) Confocal imaging of nsP1- ATTO488 (green) binding to giant unilamellar vesicles (GUVs) (red) with POPC, or POPC including 5 mol% PI(4,5)P$_2$, or 20% POPS. Scale bar, 20 μm. (**E**) Quantification of nsP1-bound GUVs from three experiment series. Data represent the percentage of nsP1-binding GUVs calculated from total number of GUVs observed for each experiment series plotted against the respective GUVs types. (**F**) Co-pelletation assay of nsP2 and nsP1 with POPS-containing MLVs. NsP2 and MLV concentrations were kept constant, while the nsP1 concentration was varied. Analysis of supernatant (**S**) and pellet (**P**) fractions by SDS-PAGE. (**G**) Quantification of pelleted nsP2 with nsP1. The experiment shown in (**F**) was repeated two times. The pellet intensity at each nsP1 concentration was normalized to the total nsP2 intensity and plotted (mean ± SD) against the nsP1 concentration. (**H**) Schematic of the findings from **A to G**, in the context of the neck complex. The non-structural proteins nsP3 and nsP4 were not included in these experiments but are displayed for completion as possible components of the neck complex.

The online version of this article includes the following source data and figure supplement(s) for figure 3:

**Source data 1.** Contains the uncropped version of the gel image shown in *Figure 3F*.

**Figure supplement 1.** The purified chikungunya virus (CHIKV) nsPs are homogeneous and monomeric.

**Figure supplement 1—source data 1.** Contains the uncropped version of the gel images shown in *Figure 3—figure supplement 1*.

**Figure supplement 2.** Representative gels related to *Figure 3*.

**Figure supplement 2—source data 1.** Contains the uncropped version of the gel images shown in *Figure 3—figure supplement 2*.

*supplement 2*). We thus concluded that the filament tracing can be used to estimate the amount of genetic material present in a single spherule. Two tomograms of sufficiently high quality, recorded on different cells and each containing a cluster of full-sized spherules ($n_1$=15 and $n_2$=6), were analyzed. The total length of filaments for each dataset were 18,600±2900 Å/spherule and 21,400±1600 Å/spherule (*Figure 4B*). Assuming that the RNA was double-stranded and adopted an A conformation, the distance between two base pairs is 2.56 Å (*Hardison and Chu, 2021*; *Tolokh et al., 2014*). Based on that assumption, the filament length corresponds to 7300±1150 and 8400±600 bp/spherule in the two tomograms, respectively (*Figure 4C*). This corresponds to an average ~80–90% of a single replicon RNA copy at 8820 bp. It thus seems parsimonious to assume that all spherules, in fact, contain exactly one full-length copy of the template strand, of which a high but variable fraction is present in the dsRNA form.

## The force exerted by RNA polymerization is sufficient to drive spherule membrane remodeling

Proteins are known to induce membrane budding when they form spherical scaffolds that template the membrane shape (*Alimohamadi and Rangamani, 2018*; *Idema and Kraft, 2019*; *Penič et al., 2020*). Since we observed viral proteins only at the spherule neck, we reasoned that other mechanisms may be involved in generating the characteristic high-curvature spherule membrane bud. In principle, the recruitment of certain lipids by nsP1 may stabilize the membrane neck, which for the similarly shaped caveolae has been shown to have a negative mean curvature (*Parton et al., 2020*). However, the in vitro reconstitution showed that nsP1 primarily binds monovalent anionic lipids (*Figure 3A–E*). Such lipids are reported to have near-zero or slightly positive spontaneous curvature, which excludes nsP1-induced lipid recruitment as a mechanism for stabilizing the spherule shape (*Dymond, 2021*). Instead, we made the biological *Ansatz* that membrane remodeling is driven by the generation of dsRNA. This is the process by which the incoming positive-strand RNA is copied into a negative strand, which will be present in a duplex with the positive strand. This process may happen in two ways: (i) the initial positive strand is present in a nascent spherule which grows as the single strand is turned into dsRNA or (ii) the initial positive strand is translocated into the spherule lumen concomitant with the production of the complementary negative strand. Either of (i) or (ii) are compatible with the model described below. We developed a mathematical model of spherule membrane shape with the following physical assumptions. We describe the membrane as a thin elastic sheet in a Helfrich-type model. The RNA, which is modeled as a semiflexible polymer, exerts a pressure onto the membrane that causes the spherule to expand. During this process the neck complex keeps the membrane neck size constant. As the dsRNA is produced, it exerts a pressure $P$ on the spherule membrane. The pressure that acts to increase spherule volume is balanced by the elastic membrane properties. To model the formation of a spherule, we begin by formulating the membrane energy $E$, which includes the Helfrich bending energy, the membrane tension $\sigma$, and the pressure $P$ exerted by the viral RNA (*Helfrich, 1973*).

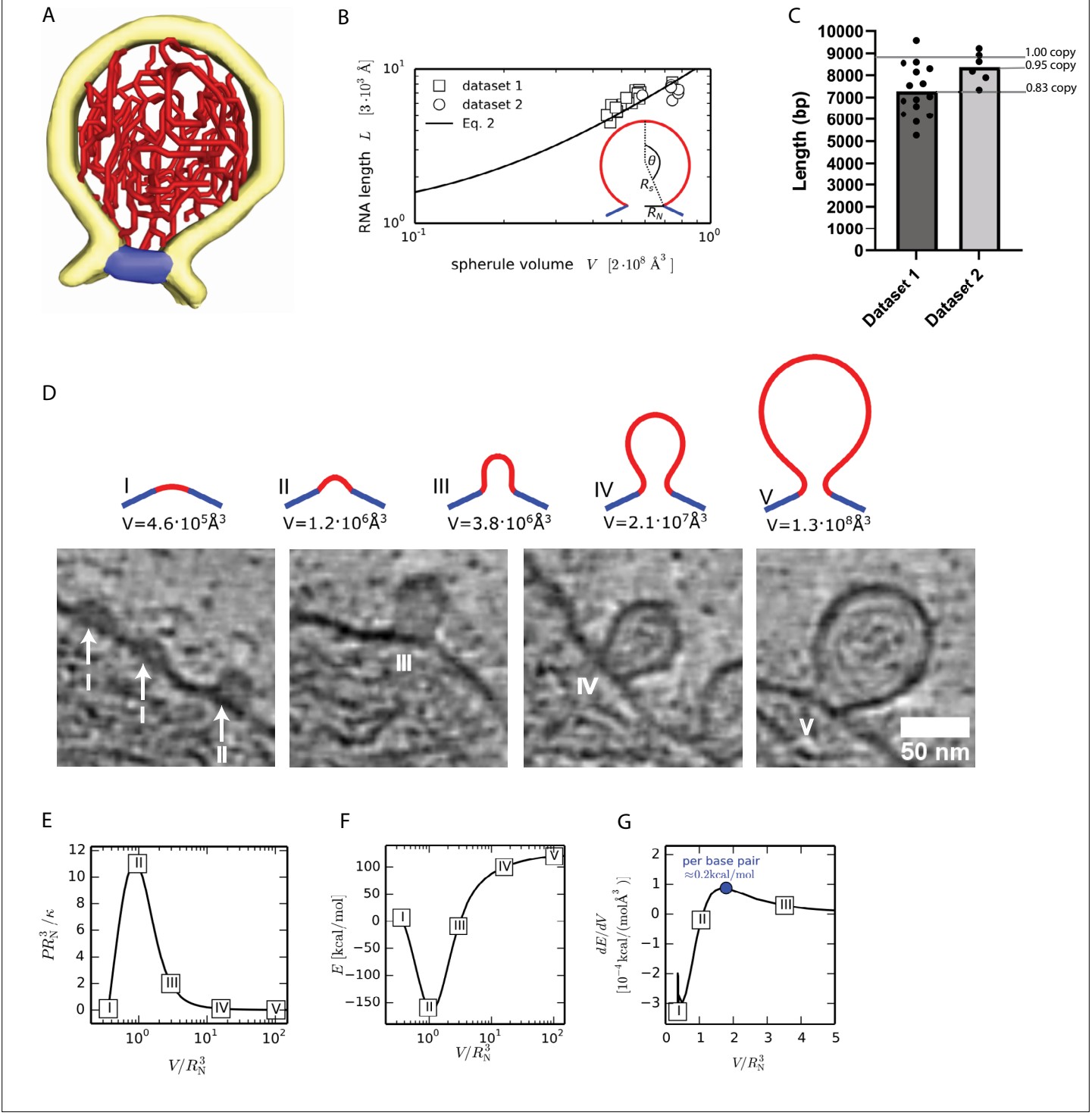

**Figure 4.** A single copy of the genomic RNA determines the shape of the spherule membrane. (**A**) Segmentation of the dsRNA traced within a spherule. Yellow: membrane, red: RNA, and blue: neck complex base. (**B**) The RNA length L increases with spherule volume V. A common fit of both datasets with **Equation 2** gives $L_0 = (3 \pm 1) \cdot 10^3 \text{Å}$ and $\sigma R_N^2/\kappa = (4 \pm 2) \cdot 10^{-2}$, while $R_N = 96\,\text{Å}$ was determined experimentally. The inset shows the spherical cap model schematically. (**C**) Estimation of the dsRNA length (in base pairs) and the average copy number per spherule. One point represents a single spherule, and the datasets represents tomograms acquired on different cells. (**D**) The top row shows five shapes that minimize the energy (**Equation 1**) for a given spherule volume. Below, the predicted shapes are compared to different sizes of nascent and full-size spherules observed in cryo-electron tomograms of Semliki forest virus (SFV) viral replicon particle (VRP)-transduced cells. Scale bar, 50 nm. (**E**) Pressure-volume relation for a unitless membrane tension of a $\sigma R_N^2/\kappa = 10^{-2}$. The corresponding membrane shapes are shown in subfigure D. (**F**) Energy (**Equation 1**) as a function

*Figure 4 continued on next page*

*Figure 4 continued*

of the spherule volume for $\sigma R_N^2/\kappa = 10^{-2}$, $\kappa=10\ k_B T$ and $R_N = 96\ \mathring{A}$. (**G**) The energy change per change in volume is shown, which leads to a maximal energy to be supplied per base pair of 0.2 kcal/mol, where we assumed the volume of a single base pair to be not larger than $2 \cdot 10^3\ \mathring{A}^3$.

The online version of this article includes the following figure supplement(s) for figure 4:

**Figure supplement 1.** Tracing of RNA in the spherule lumen and membrane neck diameter.

**Figure supplement 2.** Dependence of the total filament length on filament tracing parameters.

**Figure supplement 3.** Spherule shape parameterization.

---

$$E = \int dA \left( 2\kappa H^2 + \sigma \right) - P \int dV, \qquad (1)$$

with the membrane area $A$, the bending rigidity $\kappa$, the mean curvature $H$, and the spherule volume $V$. We solve *Equation 1* numerically (see Materials and methods and Theory section for details) for different spherule volumes to mimic the different stages of the membrane remodeling process, while fixing the area (or volume) would lead additional constraints on the solution (*Iglic and Hägerstrand, 1999*; *Hägerstrand et al., 1999*). Since the neck shape is fixed, the membrane does not go through any topological change as the spherule grows. As such, the integral over the Gaussian curvature only adds a constant to the energy, following from Gauss-Bonnet theorem, and is therefore omitted in *Equation 1*.

To derive a scaling relation between the RNA length $L$ and the spherule volume $V$, we approximate the spherule shape by a spherical cap, as indicated in the inset of *Figure 4B*, where the radius $R_s$ and the polar angle $\theta$ are related via the neck radius $R_N =R_s sin(\theta)$. In the limit of a large spherule ($\theta\approx\pi$), we find $P\sim\kappa\ (\pi-\theta)^5+2\sigma\ R_N^3\ (\pi-\theta)$ and $V\sim(\pi-\theta)^{-3}$ (see Materials and methods and Theory section). It is known from polymer theory that the pressure volume relation of long semiflexible polymers in spherical confinement scales to leading order as $PV\sim LV^{-2/3}$ (*Chen, 2016*; *Edwards and Freed, 1969*; *Morrison and Thirumalai, 2009*). Hence, the RNA length scales with the spherule volume as a power law with

$$L = L_0 \left[ 1 + \frac{\sigma R_N^2}{\kappa} 2 \left( \frac{3}{4\pi} \right)^{4/3} \left( \frac{V}{R_N^3} \right)^{4/3} \right] \qquad (2)$$

The neck radius is determined from EM imaging with $R_N = 96\mathring{A}$. Based on the data shown in *Figure 4B*, we fit a value of $L_0 = (3 \pm 1) \cdot 10^3 \mathring{A}$ for the prefactor in *Equation 2* and a scaled membrane tension $\sigma R_N^2/\kappa = (4 \pm 2) \cdot 10^{-2}$. For comparison, with a bending rigidity of $\kappa=10\ k_B T$, we obtain $\sigma=10^{-5}$ N/m, within the range of experimentally measured membrane tensions (*Baumgart et al., 2003*; *Roux et al., 2005*). As noted above, we interpret the variable length of filament traced in individual spherules as a variable (but high) fraction of the negative strand template being present in dsRNA form. The single-stranded fraction would contribute substantially less to the internal pressure by virtue of its smaller volume and much shorter persistence length and is thus for simplification ignored when fitting in *Figure 4B*. Due to the limited signal-to-noise in the tomograms, it is possible that the experimentally measured RNA length underestimates the actual RNA length slightly. For the comparison between theory and experiment, this means that the constant $L_0$ is also correspondingly smaller. An important note is that $L_0$ is not used further in the analysis. Instead, membrane tension σ plays an essential role in the theoretical model and is estimated by the slope of the data points in *Figure 4B*. A systematic underestimation of the RNA length would shift the curve slightly, but not change its slope, and thus would not influence the predicted σ.

Next, we study the membrane shape transformation from an initial pit to a fully formed spherule. The energy (*Equation 1*) is minimized using the Euler-Lagrange method (see Materials and methods and Theory section). To this end, we apply the arc-length parameterization and constrain the membrane in the neck region to the experimental geometry of the neck complex. We suppose that the shape of the membrane neck is predominantly determined by the structure of the neck complex. In principle, various other mechanisms may aid in stabilizing the membrane neck, such as the addition of anisotropic membrane inclusions (*Kralj-Iglič et al., 1999*). *Figure 4D* (upper half) shows the series of predicted membrane shapes. Notably, we were able to observe spherules at different stages in the growth process corresponding to each of the predicted membrane shapes, in cryo-electron

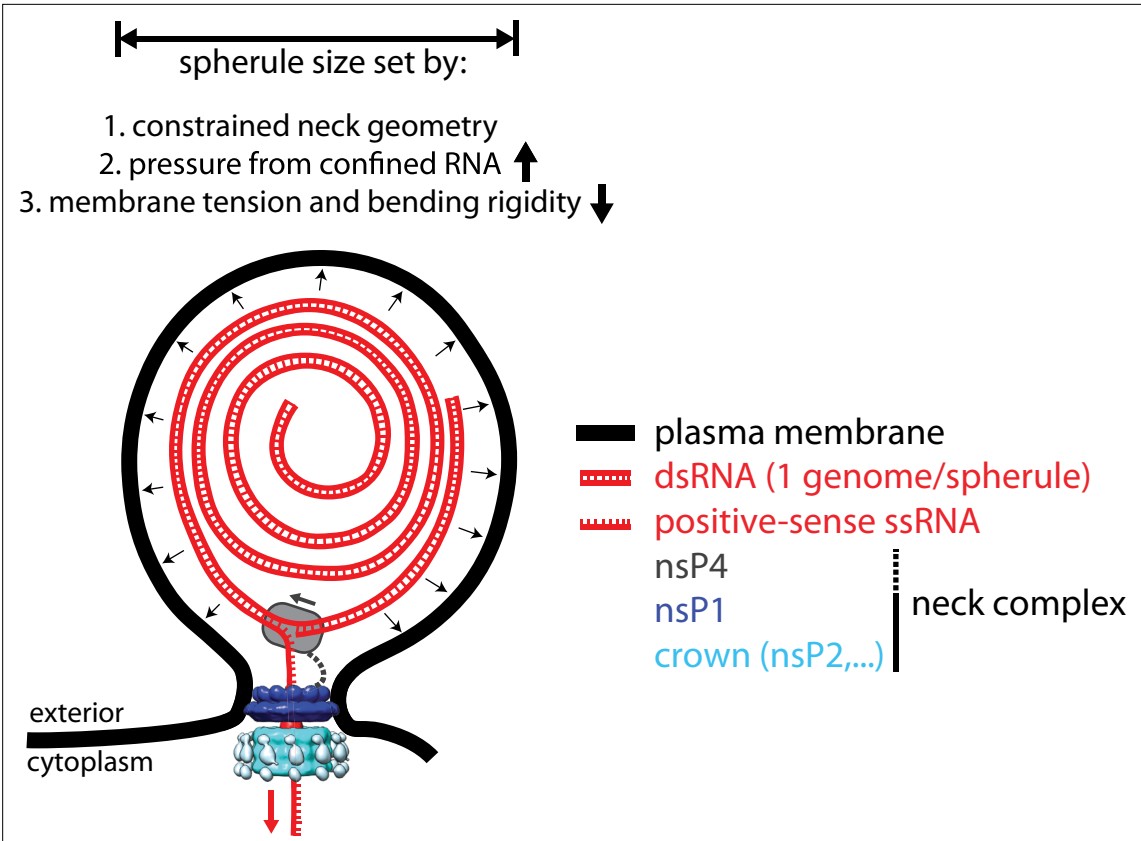

**Figure 5.** A model for the interplay between membrane, RNA, and proteins in the chikungunya virus (CHIKV) spherule. Each spherule contains a single viral genome, to >80% present in dsRNA form. The membrane shape is determined by the confined neck geometry, the pressure exerted by the confined genome, and the tension and stiffness of the membrane. NsP1 determines the neck geometry and serves an base plate for assembly of an additional 1.2 MDa complex. Biochemical evidence indicates that nsP2 is part of the neck complex, and an association of the viral polymerase nsP4 with the neck complex, although not directly addressed in this article, would be consistent with the suggested strand-displacement replication mode that produces several positive-sense strands from each spherule.

tomograms of cells transduced with replicon particles from the *Alphavirus* SFV, a close relative of CHIKV (*Figure 4D*, lower half). In *Figure 4E*, the pressure-volume relation is shown, for $\sigma R_N^2/\kappa = 10^{-2}$. We find that the largest pressure is exerted for a rather small membrane pit with a volume of $V \approx R_N^3 \approx 1.3 \cdot 10^6 \mathring{A}^3$. With a bending rigidity of $\kappa = 10\ k_B T$, we find by solving *Equation 1* that an energy barrier of roughly 250 kcal/mol has to be overcome going form a flat membrane to a fully formed spherule (*Figure 4F*). However, the energy cost per RNA base pair is much smaller. In *Figure 4G*, the change in energy per change in volume in shown. We see a maximum of $\frac{dE}{dV} \approx 1 \cdot 10^{-4}\ kcal/\left(mol \cdot \mathring{A}^3\right)$ around $V \approx 1.5 R_N^3$. Assuming that each additional RNA base pair increases the volume by at most $2 \cdot 10^3\ \mathring{A}^3$, we estimate the maximum energy to be supplied per base pair at $0.2\ kcal/mol$ at 25°C. The free energy change of RNA polymerization, including hydrogen bonding with the template, amounts to $\Delta G_0 = -1.9\ kcal/\left(mol \cdot base\right)$ without accounting for the hydrolysis of the pyrophosphate. Comparing the two, we conclude that the free energy released by RNA polymerization is around 10 times larger than the energy required to bend the membrane, even at its peak 'resistance.' Thus, RNA polymerization is sufficient to remodel the spherule membrane into its characteristic shape, assuming the neck geometry is constrained.

## Discussion

In this study, we investigated the structural organization of spherules, which are the RNA replication organelles of *Alphaviruses*. Our main findings are summarized in *Figure 5*. Four viral proteins, nsP1–nsP4, are involved in the *Alphavirus* genome replication (*Ahola et al., 2021*). High-resolution

structures have been determined for isolated domains of several nsPs, and for a ring-shaped dodecamer of the capping enzyme nsP1, the only nsP known to have membrane affinity (*Law et al., 2019*; *Shin et al., 2012*; *Jones et al., 2021*; *Zhang et al., 2021*; *Malet et al., 2009*; *Tan et al., 2022*). These structures have provided insights into individual viral enzymatic functions, but not their cellular structural context, i.e., the spherules. *Alphaviruses* are not only a major source of morbidity, but their unique RNA replication mechanism is also used to develop self-replicating RNA vaccines that induce a more potent immune response than conventional mRNA vaccines (*Ballesteros-Briones et al., 2020*). Underlying both the pathogenic viruses and the self-replicating RNA vaccine candidates is the same spherule machinery, which highlights the importance of understanding its organization. Our subtomogram average of the spherule neck complex (*Figure 2*) provides first insights into this and suggests that the ring-shaped nsP1 dodecamer serves as the assembly hub for a larger protein complex sitting at the neck of the membrane bud. In a complementary approach, we showed by in vitro reconstitution that nsP1 is necessary and sufficient for membrane association of the helicase-protease nsP2, which has no membrane affinity on its own (*Figure 3*). Hence, the in vitro reconstitution validated the tomography-based conclusion that nsP1 serves the role as neck complex assembly hub.

Our data align *Alphaviruses* with an emerging theme in positive-sense RNA virus replication: macromolecular complexes located at a membrane neck play key roles in genome replication. All positive-sense RNA viruses utilize cytoplasmic membranes to compartmentalize their RNA replication machineries. It has been suggested to split the replication compartments into two groups based on membrane topology: double-membrane vesicles and membrane buds. This grouping based on membrane topology has recently been shown to correlate with the newly proposed phyla *Pisuviricota* and *Kitrinoviricota*, respectively (*Ahola and Racaniello, 2019*). *Alphavirus* spherules fall into the *Kitrinoviricota* category together with, e.g., *Flaviviruses* and *Nodaviruses*. Whereas nothing is yet known about any *Flavivirus* neck complex, cryo-electron tomography has revealed that *Nodaviruses* have a neck complex of similar dimensions as the *Alphavirus* neck complex that we present here (*Unchwaniwala et al., 2021*; *Unchwaniwala et al., 2020*). In the *Pisuviricota* phylum, the presence of a neck complex seems less conserved. The double-membrane vesicles-type replication organelles of *Coronaviruses* indeed have a neck complex connecting their lumen with the cytosol (*Wolff et al., 2020*). On the other hand, the similarly shaped enteroviruses organelles are of autophagic origin, frequently open, and appear to lack a neck complex (*Dahmane et al., 2022*). The degree of structural conservation between neck complexes remains to be determined, but where they appear they have likely all evolved to solve the same problem: creating an environment conducive to viral genome replication in a cytoplasm rife with antiviral defense systems.

The limited resolution of the subtomogram average prevented us from determining if all enzymatic functions needed for RNA replication are localized directly in the neck complex. Conceptually, the localization of the polymerase nsP4 to the neck complex would be easy to reconcile with a strand-displacement mechanism that couples RNA polymerization to the extrusion of the displaced strand through the neck complex into the cytoplasm (*Figure 5*). An eventual high-resolution structure of the entire neck complex may resolve this question, but a complete structural understanding of *Alphavirus* RNA replication is likely to require several such structures due to the existence of a distinct early 'negative-strand' complex (*Shirako and Strauss, 1994*).

Analysis of the cryo-electron tomograms gave a clear answer to the question of the membrane bud contents: the lumen of full-size spherules contains a single copy of the viral genome in dsRNA form. While it has generally been speculated that the lumen of bud-type replication organelles contains the replicative intermediate, the consistency of the copy number is a striking outcome of our analysis. This may also suggest an explanation for why bud-type replication organelles from different *Kitrinoviricota* viruses all tend to have a similar diameter; they all contain genomes of ~8–12 kb, which at a given density would all occupy a similar volume. Our mathematical model, consistent with the tomographic data, shows that the pressure exerted by the confined dsRNA and the restriction of the neck geometry are sufficient to generate and maintain the high-energy shape of the spherule membrane. Crucially, the model also shows that the energy released by RNA polymerization is sufficient to drive the membrane shape remodeling. This establishes polymerase-driven budding as a new membrane remodeling mechanism. Notably, even though polymerase-driven budding does not require any lipids or proteins stabilizing the domed spherule membrane, it can still be assumed to lead to curvature-dependent sorting of membrane components through curvature-composition coupling (*Bashkirov*

*et al., 2022*). This may explain data suggesting the presence of the host tetraspanin CD81 at CHIKV replication sites (*Lasswitz et al., 2022*). Future studies may determine if this mechanism is generally used in the large number of positive-sense RNA viruses with 'bud-type' replication organelles. Taken together, our study takes the first steps toward an integrated structural model of an entire viral replication organelle, suggesting a high degree of spatial coordination of proteins, RNA, and membrane components of the *Alphavirus* spherule.

# Materials and methods

**Key resources table**

| Reagent type (species) or resource | Designation | Source or reference | Identifiers | Additional information |
|---|---|---|---|---|
| Strain and strain background (chikungunya virus [CHIKV]) | CHIKV viral replicon particles (VRPs) | DOI:10.1186/1743–422X-10-235 | | The viral genome is split into three RNAs, as described in the DOI, resulting in VRPs that cause a single-cycle infection. |
| Strain and strain background (*Escherichia coli*) | BL21(DE3) | In-house stock | | Used for protein expression. |
| Strain and strain background (*E. coli*) | XL10 | In-house stock | | Used for cloning. |
| Cell line: (*Mesocricetus auratus*) | Baby hamster kidney (BHK)-21 (C-13) | ATCC | Cat: CCL-10 | Fibroblast cell line. |
| Commercial assay and kit | LookOut Mycoplasma PCR detection kit | Sigma-Aldrich | Cat: MP0035 | Outcome was negative throughout this work. |
| Recombinant DNA reagent | CHIKV nsP1 *E. coli* expression plasmid | This paper | | Cloned for CHIKV nsP1 protein expression. Synthetic gene based on GenBank: KT449801.1. Available upon request. |
| Recombinant DNA reagent | CHIKV nsP2 *E. coli* expression plasmid | This paper | | Cloned for CHIKV nsP2 protein expression. Synthetic gene based on GenBank: KT449801.1. Available upon request. |
| Chemical compound and drug | Protease Inhibitor cocktail | In-house preparation | | Benzamidine: Sigma-Aldrich Cat: B6506-25G PMSF: VWR Cat: 0754–5 G Leupeptin: Alfa Aesar Cat: 15483809 |
| Chemical compound and drug | 1-palmitoyl-2-oleoyl-sn-glycero-3-phosphocholine (POPC) | Avanti Polar lipids | Cat: 850457 C | |
| Chemical compound and drug | 1-palmitoyl-2-oleoyl-sn-glycero-3-phospho-l-serine (POPS) | Avanti Polar lipids | Cat: 840034 C | |
| Chemical compound and drug | 1-palmitoyl-2-oleoyl-sn-glycero-3-phospho-(1'-rac-glycerol) (POPG) | Avanti Polar lipids | Cat: 840457 C | |
| Chemical compound and drug | Phosphatidylinositol (PI) | Avanti Polar lipids | Cat: 840042 C | |
| Chemical compound and drug | PI(4)P | Avanti Polar lipids | Cat: 840045 P | |
| Chemical compound and drug | PI(4,5)$P_2$ | Avanti Polar lipids | Cat: 840046 P | |
| Chemical compound and drug | Cholesterol | Avanti Polar lipids | Cat: 700100 P | |
| Chemical compound and drug | ATTO-647N-DOPE | Sigma-Aldrich | Cat: 42247–1 MG | Manufactured by ATTO-TEC. |
| Chemical compound and drug | ATTO-488 NHS | Invitrogen | Cat: 41698–1 MG-F | Manufactured by ATTO-TEC. |
| Software and algorithm | ImageLab | Bio-Rad | | Used for gel quatification. |
| Software and algorithm | Prism | Graph-Pad | | Used for generation of plots. |

*Continued on next page*

*Continued*

| Reagent type (species) or resource | Designation | Source or reference | Identifiers | Additional information |
|---|---|---|---|---|
| Software and algorithm | ImageJ | NIH | | Confocal images were processed using ImageJ. |
| Software and algorithm | DiscoverMP software | Refeyn Ltd | | For mass photometry data processing. |
| Software and algorithm | SerialEM | DOI:10.1016/j.jsb.2005.07.007 | | Data acquisition for cryo-electron tomography. |
| Software and algorithm | MotionCor2 | DOI:10.1038/nmeth.4193 | | Motion correction of cryo-ET data. |
| Software and algorithm | CTFFIND4 | DOI:10.1016/j.jsb.2015.08.008 | | CTF (contrast transfer function) correction of cryo-ET data. |
| Software and algorithm | IMOD | DOI:10.1006/jsbi.1996.0013 | | Reconstruction of cryo-electron tomograms. |
| Software and algorithm | Amira | Thermo Fisher Scientific | | Segmentation of cryo-electron tomograms and filament tracing. |
| Software and algorithm | Dynamo | DOI:10.1016/j.jsb.2011.12.017 | | Subtomogram averaging. |
| Software and algorithm | MATLAB | Mathworks, Inc. | | General image processing and running Dynamo. |
| Software and algorithm | UCSF chimera | DOI:10.1002/jcc.20084 | | Visualization and segmentation of subtomogram averages. |

## Cell culture

Baby hamster kidney (BHK) cells (*Mesocricetus auratus*) were a gift from Gerald McInerney (Karolinska Institutet). The cells were grown in an incubator at 37°C with 5% $CO_2$ in Minimum Essential Medium (MEM, Gibco) supplemented with GlutaMAX (Gibco) and 10% fetal bovine serium (FBS, Gibco). The cells' identity was not independently authenticated. The cells were regularly negatively tested for mycoplasma contamination using the LookOut Mycoplasma PCR detection kit (Sigma Aldrich).

## Viral replicon particles

The VRPs, described previously, were kindly provided by Andres Merits, Tartu (*Gläsker et al., 2013*). Briefly, the system consists of a viral genomic RNA in which structural proteins were replaced by a fluorescent protein, and the helper RNAs C and E coding, respectively, for the capsid and the E3, E2, 6 K, E1 structural proteins. The viral genomic RNA and the RNA of the helper plasmids C and E were in vitro transcribed and capped using the mMESSAGE mMACHINE SP6 Transcription kit (Thermo Fisher Scientific). Quality of the RNA was assessed on a denaturing formaldehyde-agarose gel. BHK cells were electroporated with the three RNA using the NEON electroporation system (Invitrogen). Cells were passaged 1 day prior to electroporation. Cells were then trypsinized and washed twice in PBS before being resuspended in the R resuspension buffer at a density of $10^7$ cells/ml and electroporated at 1200 V, 30 ms width, and one pulse. Electroporated cells were then resuspended in antibiotic-free MEM supplemented with 10% FBS and transferred to a T75 flask. After 48 hr, the medium containing the VRPs was harvested and spun down to remove detached cells and cell debris. The VRP-containing supernatant was aliquoted, flash-frozen on liquid nitrogen, and stored at –80°C. SFV VRPs were kindly given to us by Gerald McInerney (Karolinska Institutet).

## Sample preparation

QUANTIFOIL R 2/1 Au 300 EM grids were glow discharged for 10 min at 15 mAh, sterilized, and then set at the bottom of an IBIDI μ-Slide 8 well. Cells were seeded at 25,000 cells/well and left overnight to attach and spread on the EM grids. Cells were then transduced by swapping the cell medium for 250 μl of the CHIKV VRP suspension. Alternatively, SFV VRPs were added at a MOI (multiplicity of infection) of 40. 1 hr after transduction, SFV-transduced cells were treated with the PI 3-kinase

**Table 1.** Summary of the cryo-ET data collection parameters.

| Data collection | |
| --- | --- |
| Microscope | Titan Krios G2 |
| Acceleration voltage (keV) | 300 |
| Camera | Gatan K2 Summit |
| Nominal magnification | 33,000 |
| Energy filter | Yes, BioQuantum |
| Slit width (eV) | 20 |
| Pixel size in super-resolution mode (Å) | 2.18 |
| Defocus range (μm) | –3 to –5 |
| Tilt range (°) | –60 to +60 |
| Tilt increment (°) | 2 |
| Total dose (e$^-$/Å$^2$) | 80–120 |
| Tomograms used for analyses | 9 |

inhibitor Wortmannin (LC laboratories, Woburn, MA, USA) at a final concentration of 100 nM to inhibit the endocytosis of plasma membrane-located spherules (**Spuul et al., 2010**). For both CHIKV and SFV-tranduced cells, a solution of 5 nm Protein A-coupled colloidal gold (CMC-Utrecht) was added to the grids 6 hr after transduction, after which the grids were immediately plunge frozen in liquid propane-ethane using a FEI Vitrobot.

## Cryo-electron tomography

Data collection parameters are summarized in **Table 1**. Vitrified cells were imaged using a transmission electron microscope, the FEI Titan Krios with an accelerating voltage of 300 kV, a Gatan Bioquantum LS energy filter, a K2 summit detector. Tiltseries were acquired using SerialEM software (**Mastronarde, 2005**), at a magnification of 33,000× in with a super-resolution pixel size of 2.19 Å/px. Data were gathered at the plasma membrane of infected BHK cell using either a bilateral or a dose-symmetric scheme (**Hagen et al., 2017**) at a defocus between –3 and –5 μm. Typically, the total electron dose on the specimen was between 80 and 120 electrons/Å$^2$, and samples were tilted between –60° and 60° with an increment of 2°.

## Tomogram reconstruction

Movies generated during the data acquisition were motion corrected using MotionCor2 (**Zheng et al., 2017**). Tiltseries were aligned using IMOD (**Kremer et al., 1996**) based on 5 nm gold fiducials present on the specimen. The CTF was estimated using CTFFIND4 (**Rohou and Grigorieff, 2015**) and corrected using IMOD's ctfphaseflip. The images were dose filtered (**Grant and Grigorieff, 2015**), and tomograms were generated using weighted back projection in IMOD.

## Subtomogram averaging

The subtomogram averaging was carried out as schematically indicated in **Figure 2—figure supplement 1**. 76 particles were extracted from 9 high-quality unbinned tomograms using Dynamo (**Castaño-Díez et al., 2012**; **Castaño-Díez et al., 2017**). Of these 76 particles, 64 could be unambiguously oriented and centered manually before generating a first average of the protein neck complex. A cylindrical mask centered on the protein neck complex was created, and a second round of alignment was performed allowing for full-azimuthal rotations and limited (±30°) tilts with respect to the z axis (defined as the axis passing through the neck complex). Azimuthal angles of the particles in the crop table were then randomized in order to decrease the impact of the missing wedge, and by this process, another average was generated. This average was then used in combination with the original particle poses and a tighter cylindrical mask to obtain a third average. A custom mask was then defined on the center slice of the third average, radially symmetrized and used in a final alignment, still allowing full azimuthal rotations and limiting tilts and shifts. The final alignment was performed separately, once without symmetry and once with 12-fold rotational symmetry imposed. The resolutions were estimated to 34 Å and 28 Å for the unsymmetrized and C12-symmetrized averages, respectively, using the Gold-standard Fourier shell correlation with a threshold of 0.143.

## Creation of the segmented 3D models

The segmentation in **Figure 1B** was created by manual segmentation in Amira (Thermo Fisher Scientific). For the subtomogram average of the neck complex, symmetrized and non-symmetrized averages were first filtered to their respective resolution, and the tight mask was applied to them. A

smoothened representation of the membrane neck was generated by applying C36 symmetry to the average, masking away the neck complex and then applying a Gaussian filter. Both symmetrized and non-symmetrized averages were segmented in UCSF Chimera (*Pettersen et al., 2004*), and the membrane template and averages were superimposed. The published structure of nsP1 (pdb 6z0v; *Jones et al., 2021*) was filtered to the resolution of the average and then fitted in the density of the base of the neck complex using UCSF Chimera.

## Molecular mass estimation of crown subcomplex

The crown subcomplex was cropped out of the protein neck complex using the volume eraser function of Chimera. The volume of the cropped density was computed, and the molecular weight was estimated assuming 825 Da/nm$^3$ (*Erickson, 2009*).

## Filament tracing

Binned tomograms were filtered using a SIRT-like (simultaneous iterative reconstruction technique) filter with two iterations in IMOD and were imported in Amira where the RNA tracing was performed using its filament tracing functionality, a functionality that has been shown to allow quantification and structural analysis of filaments (*Rigort et al., 2012*; *Dimchev et al., 2021*). Single spherules were cropped from the imported tomograms, and a non-local means filter was applied to the cropped subtomograms with parameters selected to yield a clear contrast between the filament contained in the spherules and the background. A cylinder correlation was run with the filament width chosen to match dsRNA. The interior of spherules was segmented in order to leave out spurious hits in membranes and the exterior. Correlation lines were then traced with parameters selected to yield a good match between traces and visible filaments. The total filament length (in Å) as stated by the software was used to calculate dsRNA length in base pairs, assuming 2.56 Å/bp.

## Plasmids for protein production

Plasmids for CHIKV nsp1 and nsP2 were obtained by cloning codon-optimized CHIKV nsP1 and nsP2 genes of LR 2006_OPY1 strain into 2Bc-T vector (ORF-TEV-His6) and 1 M vector (His6-MBP-TEV-ORF), respectively, from Macrolab (University of California, Berkeley, USA).

## Expression and purification of CHIKV nsP1

To overexpress CHIKV nsP1, nsP1 plasmid was transformed into *Escherichia coli* BL21(DE3) cells. An overnight culture was grown in Luria Broth (LB) supplemented with 100 µg/ml of carbenicillin at 37°C to inoculate the secondary culture. Cells were grown at 37°C to an O.D$_{600}$ of 0.4, then the incubator temperature was reduced to 20°C. After the culture cooled down to 20°C and O.D$_{600}$ reaches between 0.8 and 0.9, protein expression was induced with 0.5 mM isopropyl β- d-1-thiogalactopyranoside (IPTG) and continue the expression at 20°C overnight. Cells were harvested by centrifuging at 7000×g (6000 rpm in a JLA-8.1000 rotor, Beckman Coulter, Brea, USA) for 60 min. After discarding the supernatant, cell pellet was washed with lysis buffer (50 mM Tris-HCl, pH 7.4, 500 mM NaCl, 0.1 mM THP (tris(hydroxypropyl)phosphine), 36 µM NP40, 5 mM MgCl2, and 10% glycerol) and stored at –80°C.

The entire purification of CHIKV nsP1 was performed at 4°C (either in the cold room or on ice). Cell pellets were thawed and resuspended in lysis buffer supplemented with DNase I and protease inhibitor cocktail (in-house preparation). Homogenized suspension then passed twice through a cell disruptor (Constant System Limited, Daventry, England) at a pressure 27 kPsi. Lysed cells were centrifugated at 36,200×g (21,000 rpm in a JA-25.50 rotor, Beckman Coulter, Brea, USA) for 1 hr, and the supernatant constituting the soluble fraction was passed through a 0.22 µm syringe filter to get a clear lysate. The cleared lysate was incubated for 2 hr at 4°C on a rotating wheel with 1 ml Ni-Sepharose Fastflow resin (Cytiva) that was pre-equilibrated with lysis buffer. After incubation, lysate-resin suspension was loaded onto a 20 ml polypropylene gravity-flow column (Bio-Rad). After collecting the flow through, the protein-bound resin was washed with wash buffer (50 mM Tris-HCl, pH 7.4, 500 mM NaCl, 0.1 mM THP, 36 µM NP40, 5 mM MgCl$_2$, 10% glycerol, and 20 mM Imidazole) twice, each with 20 column volume (CV). Washed resin was resuspended in four-CVs of lysis buffer and incubated after adding TEV (Tobacco etch virus) protease (approximately 70 µg/ml; in-house preparation) for overnight at 4°C on a rotator wheel. The cleaved protein was collected as flowthrough. An additional wash with 5 ml of lysis buffer was performed to collect the residual cleaved protein. Both elutions

were pooled and further purified by affinity chromatography. After diluting by adding buffer A (50 mM Tris-HCl, pH 7.4, 100 mM NaCl, 0.1 mM THP, 36 μM NP40, 5 mM MgCl$_2$, and 10% glycerol), diluted sample was filtered using 0.22 μM syringe filter (VWR) and loaded onto a HiTrap Heparin HP 1 ml column (GE healthcare) pre-equilibrated with buffer A. Protein was eluted over a 14 CV NaCl gradient starting at 100 mM to a final 1 M NaCl. Elutions were pooled down and concentrated using Vivaspin 6 centrifugal unit with 30 kDa cut off membrane (EMD Millipore) before being loaded onto a Superdex 200 increase 10/300 GL size-exclusion column (Cytiva) that was pre-equilibrated with size-exclusion chromatography (SEC) buffer (20 mM Tris-HCl, pH 7.4, 300 mM NaCl, 0.1 mM THP, and 5% glycerol). Protein elutions corresponding to monomeric-nsP1 peak were pooled and concentrated. Aliquots were then flash froze in liquid N$_2$ and stored at –80°C.

## Expression and purification of CHIKV nsP2

Overexpression of CHIKV nsP2 was performed using LEX bioreactor in the following manner. The nsP2 plasmid was transformed into *E. coli* BL21(DE3) cells. An overnight culture was grown in LB supplemented with 50 μg/ml of kanamycin at 37°C to inoculate the secondary culture. Before going to the LEX bioreactor, Terrific Broth (48.2 g/l of TB supplemented with glycerol at 8 ml/l) was augmented with Kanamycin (50 μg/ml) and antifoaming agent (approximately 15 drops in 1.5 l media; Sigma Aldrich). The media in 2 l bottles were kept at 37°C with bubbling for approximately 45 min and then inoculated with overnight primary culture (1:100). Around the O.D$_{600}$ 0.35–0.45 changed the temperature of the bioreactor to 18°C and let the culture to cool down to 18°C. At this point, protein expression was induced with 0.5 mM IPTG and expression continued at 18°C for 18–20 hr. Cells were harvested by centrifuging at 7000×g (6000 rpm in a JLA-8.1000 rotor, Beckman Coulter, Brea, USA) for 60 min. After discarding the supernatant, cell pellet was washed with lysis buffer (50 mM Tris-HCl, pH 8.0, 500 mM NaCl, 10% glycerol, 0.1 mM THP, and 36 μM NP-40) and stored at –80°C. The entire purification of CHIKV nsP2 was performed at 4°C. Cell mass was thawed and resuspended in lysis buffer supplemented with DNase I and protease inhibitor cocktail (in-house preparation). Homogenized suspension then passed twice through a cell disruptor (Constant System Limited, Daventry, England) at a pressure 27 kPsi. Lysed cells were centrifuged at 36,200×g (21,000 rpm in a JA-25.50 rotor, Beckman Coulter, Brea, USA) for 1 hr, and the supernatant constituting the soluble fraction was passed through a 0.22 μm syringe filter to get a clear lysate. The cleared lysate was incubated for 2 hr at 4°C on a rotating wheel with 1 ml Talon Fastflow resin (Cytiva) that was pre-equilibrated with lysis buffer. After incubation, lysate-resin suspension was loaded onto a 20 ml polypropylene gravity-flow column (Bio-Rad). After collecting the flow through, the protein-bound resin was washed with wash buffer (50 mM Tris-HCl, pH 8.0, 500 mM NaCl, 0.1 mM THP, 36 μM NP40, 10% glycerol, and 20 mM Imidazole) thrice, each with 20 CV. Protein was eluted with elution buffer (50 mM Tris-HCl, pH 8.0, 500 mM NaCl, 0.1 mM THP, 36 μM NP40, 10% glycerol, and 250 mM Imidazole) in two fractions each of 5 ml. 6xHis-tag was removed by adding TEV protease (approximately 70 μg/ml; in-house preparation) for overnight at 4°C on a rotator wheel. The cleavage mixture was centrifuged in a tabletop centrifuge at 1500×g at 4°C for 45 min to remove the visible precipitation. The supernatant was filtered using a 0.22 μM syringe filter and then pass through a HiTrap MBP-1 ml column pre-equilibrated with elution buffer to get rid of the His-MBP and His-TEV. The flowthrough, after diluting with buffer A (50 mM Tris-HCl, pH 8.0, 50 mM NaCl, 10% glycerol, and 0.1 mM THP), was filtered using 0.22 μM syringe filter and loaded onto a HiTrap Heparin HP 1 ml column (GE healthcare) pre-equilibrated with buffer A. Protein was eluted over a 14 CV NaCl gradient starting at 100 mM to a final 1 M NaCl. Elutions were pooled down and concentrated using Vivaspin 6 centrifugal unit with 30 kDa cut off membrane (EMD Millipore) before being loaded onto a Superdex 200 increase 10/300 GL size-exclusion column (Cytiva) that was pre-equilibrated with SEC buffer (50 mM HEPES-NaOH, pH 8.0, 300 mM NaCl, 10% glycerol, and 0.1 mM THP). Protein elutions corresponding to nsP2 peak were pooled and concentrated. Aliquots were then flash froze in liquid N2 and stored at –80°C.

## Fluorophore labeling of CHIKV nsP1

Fluorophore labeling was performed on the eluent of the Heparin affinity chromatography. For labeling, the purification of nsP1 from metal-based affinity chromatography to Heparin affinity chromatography was performed in same buffers, but Tris-HCl was replaced with HEPES-NaOH. The CHIKV nsP1 was mixed with threefold molar-excess of ATTO488 NHS (ATTO-TEK) and incubated at room

temperature for 2 hr. The free dye in the reaction was quenched by adding 1 M Tris-Cl, pH 7.4 to a final concentration of 50–100 mM and incubated further for 15–30 min. The CHIKV nsP1 labeling reaction was run through the HiLoad 16/600 Superdex 200 pg column pre-equilibrated with SEC buffer (20 mM Tris-HCl, pH 7.4, 300 mM NaCl, 0.1 mM THP, and 5% glycerol) to separate the monodisperse state of the labeled protein from the free dye. Labeling efficiencies were normally 70–100%.

## Liposome preparation

The phospholipids for liposome preparation, 1-palmitoyl-2-oleoyl-sn-glycero-3-phospho-l-serine (POPS), 1-palmitoyl-2-oleoyl-sn-glycero-3-phosphocholine (POPC), 1-palmitoyl-2-oleoyl-sn-glycero-3-phospho-(1'-rac-glycerol) (POPG), L-α-PI(Liver, Bovine), L-α-PI-4-phosphate (PI(4)P)(Brain, Porcine), L-α-PI-4,5-bisphosphate (PI(4,5)P$_2$) (Brain, Porcine), were purchased as chloroform (or chloroform:methanol:water) solutions, except cholesterol which was purchased as a solid and dissolved in chloroform. All lipids were p by Avanti Polar Lipids Inc.

## Multilamellar vesicles

MLVs were prepared by mixing phospholipids dissolved in solvent at the desired molar ratio (see *Table 2*). POPC was the bulk lipid, cholesterol was kept fixed at 20 mol%, and charged lipids were added to the desired percentage. Lipids with net charge <−1 were added so as to the overall charge density the same as with the corresponding MLVs with (−1) charged lipids. Chloroform was evaporated under a gentle stream of dry nitrogen gas. The dried lipid mixtures were left under vacuum overnight to completely remove all traces of chloroform and then hydrated with buffer (20 mM Tris-HCl pH 7.4, and 0.1 mM THP) to a final lipid concentration of 2 mg/ml.

## Giant unilamellar vesicles

GUVs were prepared as described previously (*Carlson and Hurley, 2012*). Briefly, a lipid mix was spread on the conductive side of the indium-tin oxide (ITO)-coated glass and left under vacuum overnight to remove all traces of chloroform. Electroformation was then performed in 600 mM sucrose solution for 1 hr at 45°C at 1 V, 10 Hz. All lipid mixes included cholesterol at 20 mol%, Atto647N-DOPE at 0.1 mol%, and POPC as bulk lipid. POPS was included at 20 mol% and PI(4,5)P$_2$ at 5 mol% to give the same nominal charge density on the membranes. To prevent segregation of PI(4,5)P$_2$ from other lipids, the lipid mix and ITO-coated glass slide were preheated to 60°C prior to spreading the lipids on the slides, and the electroformation was in this case performed for 1 hr at 60°C.

## MLVs pulldown assay

CHIKV nsP1 in SEC buffer was added to MLVs suspension in 1:1 volume ratio keeping the final lipid concentration in the mixtures at 1 mg/ml. The lipid to protein molar ratio was kept at 500:1. The mixture was incubated at room temperature for 30 min and then centrifuged at 21,130×g for 30 min at 4°C. The supernatant was carefully removed, after which equal amounts supernatant and pellet were run on 10% SDS-PAGE. After destaining the Coomassie stained gel, image was acquired with a Chemidoc Imaging System (Bio-Rad), and the relative intensity of bands were quantified using ImageLab software (Bio-Rad). Each experiment was repeated three times. Relative pellet intensity was used to calculate the MLVs bound-protein fraction and mean ± SD was plotted using Prism (Graph-Pad).

## Confocal imaging

In a Lab-Tek II chambered coverglass (Fisher Scientific), 150 µl of GUVs were mixed with 150 µl of isosmotic buffer (20 mM Tris-HCl, pH 7.4, 300 mM NaCl, and 0.1 mM THP) containing proteins at concentrations stated in Results. The mix was gently stirred and incubated 10 min at room temperature before imaging. Images were acquired using a Nikon A1R series confocal microscope equipped with a GaAsP detector and a Plan-Apochromat 60× oil (N.A 1.40) DIC objective. The ATTO647–DOPE membrane marker and the ATTO488–labeled nsP1 were excited with 633-, and 488 nm lasers, respectively. Z stacks of GUVs were acquired at positions selected without observing the fluorescence channels. Each stack consisting of 10 images, spaced at 1 µm. Three experiment series were performed on three separate occasions with different batches of GUVs. For each series, images were acquired from total three wells, and from each well GUVs were imaged from 10 different field views. In each set of z stack, the nsP1 binding was calculated as the fraction of GUVs having visible nsP1 fluorescence above

**Table 2.** Lipid compositions used to prepare multilamellar vesicles (MLVs).

MLVs with (–1) charged lipids

| MLVs (equivalent charge %) | 1-Palmitoyl-2-oleoyl-sn-glycero-3-phospho-l-serine/1-palmitoyl-2-oleoyl-sn-glycero-3-phospho-(1'-rac-glycerol)/phosphatidylinositol (PI) (mol %) | Cholesterol (mol %) | 1-Palmitoyl-2-oleoyl-sn-glycero-3-phosphocholine (POPC) (mol %) |
|---|---|---|---|
| PS/PG/PI (0 %) | 0 | 20 | 80 |
| PS/PG/PI (20 %) | 20 | 20 | 60 |
| PS/PG/PI (40 %) | 40 | 20 | 40 |
| PS/PG/PI (60 %) | 60 | 20 | 20 |
| PS/PG/PI (80 %) | 80 | 20 | 0 |

MLVs with (–2.5) charged lipid

| MLVs (equivalent charge %) | PI(4)P (mol %) | Cholesterol (mol %) | POPC (mol %) |
|---|---|---|---|
| PI(4)P (0 %) | 0 | 20 | 80 |
| PI(4)P (20 %) | 8 | 20 | 72 |
| PI(4)P (40 %) | 16 | 20 | 64 |
| PI(4)P (60 %) | 24 | 20 | 56 |
| PI(4)P (80 %) | 32 | 20 | 48 |

MLVs with (–4) charged lipid

| MLVs (equivalent charge %) | PI(4,5)P$_2$ (mol %) | Cholesterol (mol %) | POPC (mol %) |
|---|---|---|---|
| PI(4,5)P$_2$ (0 %) | 0 | 20 | 80 |
| PI(4,5)P$_2$ (20 %) | 5 | 20 | 75 |
| PI(4,5)P$_2$ (40 %) | 10 | 20 | 70 |
| PI(4,5)P$_2$ (60 %) | 15 | 20 | 65 |
| PI(4,5)P$_2$ (80 %) | 20 | 20 | 60 |

background. Data from all the three experiment series were then plotted against the respective GUVs types using Prism (Graph-Pad).

## nsP1-nsP2-membrane co-pelletation assay

MLVs of the lipid compositions POPC (10 %): cholesterol (20 %): POPS (70 %) were used. In this assay, we kept the nsP2 concentration fixed to 0.55 µm and nsP1 concentration was titrated from 0 µM to 2.75 µM, i.e., 1:5 in molar ratio. The final lipid concentration was kept at 1 mg/ml. The assay was performed as described above for MLV pulldown assay. The resulting gel was then silver stained.

Images were acquired with a Chemidoc Imaging System (Bio-Rad), and the relative intensity of bands was quantified using ImageLab software (Bio-Rad). Each experiment was repeated two times. The pellet intensity at each nsP1 concentration was normalized to the total nsP2 intensity and plotted (mean ± SD) against the nsP1 concentration using Prism (Graph-Pad).

## Mass photometry

Mass photometry (MP) measurement was performed on a Refeyn OneMP (Refeyn Ltd.). Microscope coverslips (24 mm × 50 mm; Paul Marienfeld GmbH) were cleaned by serial rinsing with Milli-Q water and HPLC-grade isopropanol (Fisher Scientific Ltd.), on which a CultureWell gasket (Grace Biolabs) was then placed. For each measurement, 16 µl of SEC buffer (20 mM Tris-HCl, pH 7.4, 300 mM NaCl, 0.1 mM THP, and 5% glycerol) was placed in the well for focusing, after which 4 µl of nsP1 protein was added and mixed. The final protein concentration was 5 nM. Movies were recorded for 60 s at 100 fps under standard settings. Before measuring the protein sample, a protein standard mixture was measured to obtain a standard molecular weight calibration file. Data was processed using DiscoverMP software (Refeyn Ltd).

## Theory

### 1. Predicting the relation between RNA length and spherule volume

To derive a scaling relation between the length of the RNA and the volume of the spherule, we describe the spherule shape as a spherical cap (see inset in *Figure 4B*), where the radius $R_s$ and the polar angle $\theta$ are related via the neck radius $R_N = R_s \sin(\theta)$. The volume and area are then given by $V = \frac{\pi}{3} R_N^3 \frac{(2+\cos\theta)(1-\cos\theta)^2}{\sin\theta^3}$ and $A = 2\pi R_N^2 \frac{1-\cos\theta}{\sin\theta^2}$ and the membrane energy (*Equation 1* in the Results section) reads

$$\frac{F}{\pi\kappa} = 4(1-x) + 2\frac{\sigma R_N^2}{\kappa}\frac{1-x}{1-x^2} - \frac{PR_N^3}{\kappa}\frac{(2+x)(1-x)^2}{3(1-x^2)^{\frac{3}{2}}}, \tag{T1}$$

with $x = \cos\theta$. Minimization with respect to $x$ leads to

$$\frac{dF/(\pi\kappa)}{dx} = -4 - \frac{\sigma R_N^2}{\kappa}\frac{2}{(1+x)^2} + \frac{PR_N^3}{\kappa}\frac{1}{(1+x)^2\sqrt{1-x^2}} = 0 \tag{T2}$$

and

$$\frac{PR_N^3}{\kappa} = 4(1+\cos\theta)^2\sin\theta + 2\frac{\sigma R_N^2}{\kappa}\sin\theta. \tag{T3}$$

For a fully formed spherule, i.e., $\theta \approx \pi$, we write the pressure (*Equation T3*) as a Taylor expansion around $\theta = \pi$:

$$\frac{PR_N^3}{\kappa} = (\pi-\theta)^5 + \frac{\sigma R_N^2}{\kappa}\left[2(\pi-\theta) - \frac{1}{3}(\pi-\theta)^3 + \frac{1}{60}(\pi-\theta)^5\right] + \mathcal{O}\left((\pi-\theta)^6\right) \tag{T4}$$

In analogy, the inverse of the spherule volume is expressed as a Taylor expansion around $\theta = \pi$:

$$\frac{R_N^3}{V} = \frac{3}{4\pi}(\pi-\theta)^3 + \mathcal{O}\left((\pi-\theta)^4\right) \tag{T5}$$

Inserting *Equation T5* into *Equation T4* we find:

$$P \approx \frac{\kappa}{R_N^3}\left(\frac{4\pi}{3}\right)^{\frac{5}{3}}\left(\frac{V}{R_N^3}\right)^{-\frac{5}{3}} + \frac{\sigma}{R_N}\left[2\left(\frac{4\pi}{3}\right)^{\frac{1}{3}}\left(\frac{V}{R_N^3}\right)^{-\frac{1}{3}} - \frac{1}{3}\left(\frac{4\pi}{3}\right)\left(\frac{V}{R_N^3}\right)^{-1} + \frac{1}{60}\left(\frac{4\pi}{3}\right)^{\frac{5}{3}}\left(\frac{V}{R_N^3}\right)^{-\frac{5}{3}}\right] \tag{T6}$$

Since $V \gg R_N^3$ for a mature spherule, the contribution to *Equation T6* that scale with the membrane tension are dominated by the $\left(\frac{V}{R_N^3}\right)^{-1/3}$ term. Hence, *Equation T6* simplifies to

$$P \approx \frac{\kappa}{R_N^3}\left(\frac{4\pi}{3}\right)^{\frac{5}{3}}\left(\frac{V}{R_N^3}\right)^{-\frac{5}{3}} + \frac{\sigma}{R_N}2\left(\frac{4\pi}{3}\right)^{\frac{1}{3}}\left(\frac{V}{R_N^3}\right)^{-\frac{1}{3}}. \tag{T7}$$

From polymer theory, it is known $P$, $V$, and $L$ the RNA length, scale as $PV \sim LV^{-2/3}$, or equivalently $L \sim PV^{5/3}$ (**Chen, 2016**; **Edwards and Freed, 1969**; **Morrison and Thirumalai, 2009**). Inserting **Equation T5** and **Equation T7**, we find

$$L \sim \frac{\kappa}{R_N^3} \left(\frac{4\pi}{3}\right)^{\frac{5}{3}} \left[1 + \frac{\sigma R_N^2}{\kappa} 2 \left(\frac{4\pi}{3}\right)^{-\frac{4}{3}} \left(\frac{V}{R_N^3}\right)^{\frac{4}{3}}\right], \tag{T8}$$

which is equivalent to **Equation 2** in the Results section.

$$L = L_0 \left[1 + \frac{\sigma R_N^2}{\kappa} 2 \left(\frac{3}{4\pi}\right)^{4/3} \left(\frac{V}{R_N^3}\right)^{4/3}\right]. \tag{T9}$$

## 2. Membrane shape transformation

To study the membrane shape transformation going from a flat membrane to a full-sized spherule, we derive the shape equations based on the Euler-Lagrange formalism. To this end, we describe the membrane shape in a cylindrically symmetric shape by an arc length parameterization (**Figure 4—figure supplement 3A**) with the arc length $S$ and the azimuthal angle $\psi$. The height $Z$ and the radial coordinate $R$ are then obtained via $\frac{dR}{dS} = \cos\psi$, $\frac{dZ}{dS} = -\sin\psi$, and the principle curvatures are given by $C_1 = \frac{\sin\psi}{R}$, $C_2 = \frac{d\psi}{dS}$, with the mean curvature $H = (C_1 + C_2)/2$ (**Deserno, 2015**). The membrane energy then reads

$$F = 2\pi \int_0^{S_{end}} dS \left[\frac{\kappa}{2} R \left(\frac{d\psi}{dS} + \frac{\sin\psi}{R}\right)^2 + \sigma R - P\frac{1}{2}R^2 \sin\psi\right] \tag{T10}$$

To determine the energy minimizing shape, we consider the functional $\widetilde{F}$ with the Lagrangian-like function $\mathfrak{L}$:

$$\widetilde{F} = \int_0^{S_{end}} ds \mathfrak{L}, \quad \mathfrak{L} = r \left(\psi' + \frac{\sin\psi}{R}\right)^2 + 2\widetilde{\sigma} r - \widetilde{p} r^2 \sin\psi + \gamma \left(r' - \cos\psi\right) \tag{T11}$$

We used unitless variables $s = S/R_N$, $r = R/R_N$, $\widetilde{\sigma} = \sigma R_N^2/\kappa$, $\widetilde{p} = PR_N^3/\kappa$ and where derivatives with respect to $s$ are indicated as $\frac{d}{ds} = ()'$. Furthermore, we introduce the unitless variables $z = Z/R_N$ and $v = V/R_N^3$ which will be used further below. The Lagrange multiplier function $\gamma$ enforces the geometrical relation between $r$ and $\psi$. Using the Euler-Lagrange formalism (**Jülicher, 1994**; **Seifert et al., 1991**), we find based on $\frac{d}{ds}\frac{\partial\mathfrak{L}}{\partial\psi'} = \frac{\partial\mathfrak{L}}{\partial\psi}$

$$h' = \frac{\gamma}{4}\frac{\sin\psi}{r} - \frac{\widetilde{p}}{4}r\cos\psi, \quad \text{with} \quad h = \frac{\psi' + \frac{\sin\psi}{r}}{2} \tag{T12}$$

and based on $\frac{d}{ds}\frac{\partial\mathfrak{L}}{\partial r'} = \frac{\partial\mathfrak{L}}{\partial r}$

$$\gamma' = 4h \left(h - \frac{\sin\psi}{r}\right) + 2\widetilde{\sigma} - \widetilde{p} r \sin\psi \tag{T13}$$

The spherule geometry requires the following boundary conditions

$$r(0) = 0, \quad \psi(0) = 0, \quad r(s_{end}) = 1, \quad \psi(s_{end}) = 0.14\pi \tag{T14}$$

where the shape of the membrane neck is constrained by the protein complex to a radius $R_N$, i.e., $r(s_{end}) = 1$, and an angle $\psi = 0.14\pi$. To find a boundary condition for the Lagrange multiplier function $\gamma$, we determine the Hamiltonian-like function $\mathfrak{H}$,

$$\mathfrak{H} = -\mathfrak{L} + \psi'\frac{\partial\mathfrak{L}}{\partial\psi'} + r'\frac{\partial\mathfrak{L}}{\partial r'} = rh \left(h - \frac{sin\psi}{r}\right) - 2\sigma r + pr^2 sin\psi + \gamma cos\psi \tag{T15}$$

We note that $\mathfrak{H}$ is not an energy but rather an auxiliary function that we use to derive an additional boundary condition. The explicit and implicit dependence of $\mathfrak{H}$ and $\mathfrak{L}$ on $s$ are related as $\frac{d\mathfrak{H}}{ds} = -\frac{\partial\mathfrak{L}}{\partial s}$. Since $\mathfrak{L}$ does not depend explicitly on $s$, $\mathfrak{H}$ is constant. The upper integration boundary $s_{end}$ is not

fixed, which leads to $\mathfrak{H} = 0$ (*Jülicher, 1994*; *Seifert et al., 1991*). From *Equation T15*, we can now determine the boundary condition.

$$\gamma(0) = 0 \tag{T16}$$

In summary, we obtain the following shape equations and boundary conditions.

$$r' = \cos\psi \tag{T17a}$$

$$z' = -\sin\psi \tag{T17b}$$

$$\psi' = 2h - \frac{\sin\psi}{r} \tag{T17c}$$

$$\gamma' = 4h\left(h - \frac{\sin\psi}{r}\right) + 2\widetilde{\sigma} - \widetilde{p}r\sin\psi \tag{T17d}$$

$$h' = \frac{\gamma}{4}\frac{\sin\psi}{r} - \frac{\widetilde{p}}{4}r\cos\psi \tag{T17e}$$

$$v' = \pi r^2 \sin\psi \tag{T17f}$$

$$r(0) = 0, \ z(0) = 0, \ \psi(0) = 0, \ \gamma(0) = 0, \ v(0) = 0 \tag{T18a}$$

$$r(s_{end}) = 1, \ \psi(s_{end}) = 0.14\pi \tag{T18b}$$

Since *Equation (T16)* has a singularity for $r = 0$, we shift the inner boundary from $s = 0$ to $s = \tau$. In the numerical calculations, $\tau$ is set to $\tau = 0.001$. The mean curvature at the inner boundary is denoted as $h_0$. From $\psi(\tau) = \int_0^\tau \psi' ds \approx \int_0^\tau h_0 ds = h_0\tau$, we find the new boundary condition $\psi(\tau) = h_0\tau$. And from $r' = \cos\psi \approx 1 - \frac{\psi^2}{2} \approx 1 - \frac{(h_0\tau)^2}{2}$, we find $r(\tau) = \tau + \mathcal{O}(\tau^3)$. In analogy, we obtain the following boundary conditions at $= \tau$:

$$r(\tau) = 0, \ z(\tau) = 0, \ \psi(\tau) = h_0\tau, \ \gamma(\tau) = 0, \ h(\tau) = h_0, \ v(\tau) = 0 \tag{T19a}$$

$$r(s_{end}) = 1, \ \psi(s_{end}) = 0.14\pi \tag{T19b}$$

For a given values of $\widetilde{\sigma}$ and $\widetilde{p}$, we have to find $h_0$ and $s_{end}$, such that the shape equations in *Equation (T17)* with the boundary conditions (*Equation T19*) are fulfilled. Values for $h_0$ and $s_{end}$ as a function of $\widetilde{p}$ for $\widetilde{\sigma} = 0.01$ are shown in *Figure 4—figure supplement 3* B-C. We note that similar methods have been applied to study membrane vesicles with various area to volume ratios, where $\widetilde{\sigma}$ and $\widetilde{p}$ act as Lagrange multipliers (*Iglic and Hägerstrand, 1999*; *Hägerstrand et al., 1999*).

## Acknowledgements

We thank Andres Merits (Tartu) for sharing the CHIKV VRP system, Gerald McInerney (Karolinska Institutet) for the SFV VRPs, and Tero Ahola (Helsinki) for insightful discussions at the early stage of this project. Cryo-EM and fluorescence microscopy were performed at the Umeå Center for Electron Microscopy (UCEM) and Biochemical Imaging Center Umeå (BICU), respectively. UCEM is a SciLifeLab National Cryo-EM facility supported by instrumentation grants from the Knut and Alice Wallenberg Foundation and the Kempe Foundations. UCEM and BICU are part of the National Microscopy Infrastructure, NMI (VR-RFI 2016–00968). This project was funded by a Human Frontier Science Program Career Development Award (CDA00047/2017 C) to L-AC, the Knut and Alice Wallenberg Foundation (through the Wallenberg Center for Molecular Medicine Umeå), the Swedish research council (grants 2018–05851, 2021–01145) and a Kempe foundations postdoctoral fellowship to PK.

## Additional information

### Funding

| Funder | Grant reference number | Author |
| --- | --- | --- |
| Human Frontier Science Program | CDA00047/2017-C | Lars-Anders Carlson |

| Funder | Grant reference number | Author |
| --- | --- | --- |
| Vetenskapsrådet | 2018-05851 | Lars-Anders Carlson |
| Vetenskapsrådet | 2021-01145 | Lars-Anders Carlson |
| Kempestiftelserna | JCK-1723.2 | Pravin Kumar |
| Max Planck Institute for the Physics of Complex Systems | open access funding | Susanne Liese |

The funders had no role in study design, data collection and interpretation, or the decision to submit the work for publication.

### Author contributions

Timothée Laurent, Pravin Kumar, Susanne Liese, Conceptualization, Formal analysis, Investigation, Writing – original draft, Writing – review and editing; Farnaz Zare, Mattias Jonasson, Investigation; Andreas Carlson, Lars-Anders Carlson, Conceptualization, Formal analysis, Funding acquisition, Writing – original draft, Writing – review and editing

### Author ORCIDs

Timothée Laurent http://orcid.org/0000-0003-0174-723X
Susanne Liese http://orcid.org/0000-0001-7420-5488
Farnaz Zare http://orcid.org/0000-0001-5619-8165
Mattias Jonasson http://orcid.org/0000-0001-5528-3405
Andreas Carlson http://orcid.org/0000-0002-3068-9983
Lars-Anders Carlson http://orcid.org/0000-0003-2342-6488

### Decision letter and Author response

Decision letter https://doi.org/10.7554/eLife.83042.sa1
Author response https://doi.org/10.7554/eLife.83042.sa2

## Additional files

### Supplementary files

- MDAR checklist

### Data availability

The subtomogram averages of the neck complex have been deposited at the Electron Microscopy Data Bank with accession codes EMD-14686 (unsymmetrized) and EMD-14687 (C12-symmetrized). Two reconstructed tomograms of CHIKV spherules at the plasma membrane, binned by a factor 4, are also available with the accession codes EMD-15582 and EMD-15583.

The following datasets were generated:

| Author(s) | Year | Dataset title | Dataset URL | Database and Identifier |
| --- | --- | --- | --- | --- |
| Laurent T, Carlson LA | 2022 | Subtomogram average of the chikungunya virus neck complex, unsymmetrized | https://www.ebi.ac.uk/emdb/EMD-14686 | EMDB, EMD-14686 |
| Laurent T, Carlson LA | 2022 | Subtomogram average of chikungunya virus neck complex, C12 symmetry | https://www.ebi.ac.uk/emdb/EMD-14687 | EMDB, EMD-14687 |
| Laurent T, Carlson LA | 2022 | Cryo-electron tomogram of chikungunya virus spherules at the plasma membrane | https://www.ebi.ac.uk/emdb/EMD-15582 | EMDB, EMD-15582 |
| Laurent T, Carlson LA | 2022 | Cryo-electron tomogram of chikungunya virus spherules at the plasma membrane | https://www.ebi.ac.uk/emdb/EMD-15583 | EMDB, EMD-15583 |

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
