## [Editor Report]

Chikungunya virus is a very important human pathogen, and research on the architecture of its replication/transcription organelle holds great promise for the development of future therapies. Laurent and colleagues advanced this field by providing pioneering low-resolution 3D structures of the membrane-bound viral protein complex and the viral RNA content of this organelle in situ. In addition, they also assessed the lipid requirements for membrane interaction of the primary viral membrane anchor of this complex, nsP1, in vitro.

---

## [Decision Letter]

[Editors' note: this paper was reviewed by Review Commons.]

Thank you for submitting your article "Architecture of the chikungunya virus replication organelle" for consideration by *eLife*. Your article has been reviewed by 3 peer reviewers at Review Commons, and the evaluation at *eLife* has been overseen by a Reviewing Editor and Vivek Malhotra as the Senior Editor.

Based on your manuscript, the reviews and your responses, we invite you to submit a revised version incorporating the revisions as outlined in your response to the reviews.

When preparing your revisions, please also address the following points, still outstanding from the reviewers:

1. To strengthen the argument that the spherules contain dsRNA, the authors should add discussion of the following points:

– It has previously been shown by immunofluorescence imaging that Alphavirus replication sites contain high loads of dsRNA (https://journals.asm.org/doi/full/10.1128/JVI.00085-11).

– The viral nsP2 protein, which the authors show is part of the neck complex, has RNA helicase activity and therefore likely plays a role in unwinding the dsRNA genome for replication and transcription (https://www.sciencedirect.com/science/article/pii/S0021925820440335?via%3Dihub).

2. The following two interpretations of the same data still seem contradictory.

line 208:

"Given the limited signal-to-noise ratio of cryo-electron tomograms it seems parsimonious to assume that all spherules in fact contain exactly one full-length copy of the viral genome, of which the tracing algorithm is able to detect 80-90%."

lines 257-263:

"As noted above, the experimentally measured RNA length likely underestimates the actual RNA length by 10-20%. […] A systematic underestimation of the RNA length would shift the curve slightly, but not change its slope […]"

To the best of my understanding, I disagree: if each spherule indeed had the same nucleic acid content, as seemingly implied by the first statement, then the slope of the curve from the second statement should equal zero! This requires clarification.

---

## [Author Response]

Reviewer #1 (Evidence, reproducibility and clarity (Required)):The authors proposed that the stable and opened membrane neck that connects the bud to the cytoplasm may persist for a long time in the infected cell during active RNA production. The viral ring-shaped nsPs is supposed to have an important role of maintaining this stable high-curvature membrane neck. It is suggested that the nsP1 dodecamer may pull together the membrane inner surface in the neck region via electrostatic interactions. Namely the authors observed that in the absence of negatively charged membrane lipids nsP1 did not bind appreciably to the membrane. The presented experimental data and theoretical consideration suggest that the CHIKV spherule consists of a membrane bud filled with viral RNA, and has a macromolecular complex gating the opening of this bud to the cytoplasm.The presented results are interesting, the manuscript is well written and can be published after revision. The following comments are offered to the authors' consideration.

We thank the reviewer for this positive overall assessment.

1. Since there is no protein coating over the curved surface of the membrane bud, the authors concluded that the membrane neck must be stabilised by specific mechanism involving nsP1. It was further assumed that the viral protein nsP1 serves as a base for the assembly of of a larger protein complex at the neck of the membrane bud. In addition to suggested mechanism of the neck stabilization, thin highly curved membrane neck can be stabilised also by accumulation of the membrane components having the appropriate membrane curvature (i. E. negative intrinsic curvature or anisotropic intrinsic curvature), see Kralj-Iglic et al., Eur. Phys. J. B., 10: 5-8 (1999), https://doi.org/10.1007/s100510050822.Please discuss this issue in the manuscript.

This is a good point, thank you for making it. In the revised manuscript we discuss both the possibility of lipid sorting into the neck region by nsP1 (lines 217-222), and the mentioned paper regarding anisotropic inclusions (lines 268-271).

2. In Equation (1) the Gaussian curvature term (appearing in Helfrich bending energy term) is not included. Usually this term is omitted in the case of closed membrane shapes (i.e. so-called spherical topology) due to validity of the Gauss-Bonnet theorem. In the present manuscript/work the shape equation was solved for the membrane patch. Can you therefore please explain shortly to the reader why you can omit the Gaussian curvature term from Equation (1). For example due fixed inclination angle and foxed curvature at the boundary.

Thanks for finding this omission. We have now revised the manuscript to describe why we can omit the Gaussian curvature term (lines 241-245).

3. «Sigma« and »P« can be considered also as global Lagrange multipliers for the constraint of the fixed total membrane area of the bud (including the neck membrane) and the constraint of the fixed volume of the bud. If you then take into account separately also the equation for the fixed membrane area you could predict different shapes of the bud (by solving the shape equation) at fixed area of the bud, calculated for different values of the model parameters (and different boundary conditions) – in this case Σ is the result of variational procedure (as well P if you consider also the constraint for the fixed volume of the bud). See for example Medical and Biological Engineering and Computing, vol. 37, pp. 125-129, 1999 and J. Phys. Condens. Matter, vol. 4, pp. 1647-1657, 1992. Can you please shortly discuss in the manuscript also this issue.

This is an interesting point. We now discuss this and cite the mentioned papers at the end of the theory section in the supplementary information (lines 206-208) as well as briefly mentioning it when discussing Equation 1 (lines 240-242).

Referees cross-commentingI agree as well.Reviewer #1 (Significance (Required)):The presented experimental and theoretical results are interesting, the manuscript is well written and can be published after revision.

We thank the reviewer for this appreciative comment.

Reviewer #2 (Evidence, reproducibility and clarity (Required)):Summary:In their manuscript "Architecture of the chikungunya virus replication organelle" Laurent and colleagues show:- the 3D structure of the "neck complex" that forms the gateway between the Chikungunya virus replication/transcription organelle (termed "spherule") and the cytoplasm of infected cells. The structure was obtained by native electron cryo-tomography and sub-tomogram averaging of BHK cells infected with a single-cycle replicon system encoding all components of the viral replication machinery. The nominal resolution of the structure is 28 Å. The viral nsP1 protein, for which two high-resolution structures have previously been published, could unambiguously be located within the density of the neck complex.- nsP1 interaction with membranes relies on lipids with a single negative net charge, such as POPS, POPG and PI, whereas two different PIPs with a negative net charge greater than one support nsP1 binding less efficiently. These membrane determinants for nsP1 binding were elucidated using two complementary methods: multilamellar vesicle pulldown assays and confocal imaging of fluorescently labeled giant unilamellar vesicles in the presence of fluorescently labeled nsP1. Purified nsP1 was produced in *E. coli*.- nsP1 recruits nsP2 (another component of the neck complex) to membranes with suitable lipid composition. This observation was made using the same multilamellar vesicle pulldown assay.- the 3D organization of the viral genome within the spherule, demonstrating that each spherule contains one copy of the genome as a double-stranded RNA molecule. This analysis was carried out by segmentation of the same tomograms that were used to visualize the neck complex.- the force exerted by RNA polymerization within the spherules is sufficient to drive membrane remodeling. This is a theoretical argument based on mathematical modelling.Major comments:The article is written clearly and all major claims seem justified. The biochemical assays are presented in duplicates or triplicates, which is sufficient to derive the provided conclusions. The workflow for electron cryo-tomography analysis seems sound, even though the low number of individual particles (=64) for sub-tomogram averaging of the neck complex limits the resolution of its final structure. Given the strong competition in the field, and considering the high experimental workload that would be required for further improvement of the resolution, I do not recommend any additional benchwork for this paper.

We thank the reviewer for this assessment, especially for recognising the challenge in obtaining a larger number of spherule subtomograms under the complex replicon particle protocol we had to use in order to study the BSL3 CHIKV under BSL2 conditions.

My only concern is the accuracy of the experimental genome length measurements, which has important implications for their mechanistic interpretation. The type of tomograms that have been recorded here inherently suffers from anisotropy with respect to both resolution and contrast. This makes accurate tracing of tangled filaments very challenging, and in this light, I congratulate the authors for the impressively good agreement of their average experimentally determined genome length with the theoretical genome length (Figure 4C). As to be expected, however, the second supplementary video clearly shows multiple gaps in the traced genome, implying that there must necessarily be errors in the length measurements. Unless there is a possibility to confidently estimate the magnitude of these errors, my preferred interpretation would be that the vast majority of imaged spherules – regardless of their temporary volume in the moment of sample freezing – likely contains precisely one copy of the double-stranded RNA genome, and not fractions thereof as is suggested in the text (for example, line 305: "Analysis of the cryo-electron tomograms gave a clear answer to the question of the membrane bud contents: the lumen of full-size spherules consistently contains 0.8-0.9 copies."). I feel that this subject deserves more discussion in the manuscript. If the authors prefer to keep their original interpretation that the majority of spherules contains only fractions of full genomes, I invite them to provide an explanation for why their experimental genome length measurements are sufficiently accurate to favor this rather surprising conclusion over my more trivial interpretation. If I understand correctly, my preferred interpretation has implications for the mathematical model for membrane remodeling (Equation 2).

This is a good point. In fact, we agree that our original manuscript and wording was unclear and we agree with the reviewer’s interpretation (“my preferred interpretation would be that the vast majority of imaged spherules – regardless of their temporary volume in the moment of sample freezing – likely contains precisely one copy of the double-stranded RNA genome”). We have now changed the text to reflect that we believe we have a 10-20% false negative rate in the filament tracing and that the most likely interpretation is indeed that each spherule has exactly one genome copy (lines 207-210). In addition, we looked at the possible consequences of the slight underestimation of the filament length for the mathematical model, and describe on lines 257-264 why this in fact would have no impact on the conclusions of the modeling.

Minor comments:Virus taxa should be capitalized and written in italics wherever applicable. I recommend adhering to the following rules:https://talk.ictvonline.org/information/w/faq/386/how-to-write-virus-species-and-other-taxa-names

Thank you for helping us clarify this. In response to this we have now italicized and capitalized all virus taxa.

Figure 2I looks as if the pink cross-section of nsP1 has not been scaled correctly. Comparison to Figure 2H gives me the impression that the diameter of the pink nsP1 ring in Figure 2I should be scaled down relative to the greyscale neck complex.

We would like to than the reviewer for their keen eye. There was indeed a scaling problem, which we have now solved in the updated Figure 2.

The caption of Figure 2 calls more panels than are provided in the figure. The caption "panel E" seems to be obsolete.

Thanks for finding this mistake. We have now revised Figure 2 and its legend.

In the methods, centrifugation speed should be given in units of relative centrifugal force (rcf) instead of revolutions per minute (rpm), especially for the MLV pulldown assay where no rotor is indicated.

We agree and have changed this on lines 482,490,524,531,543 and 597 of the manuscript

In the methods for the MLV assay, the lipid:protein ratio is given with 500:1. It should be specified whether this is a mass ratio or a molar ratio.

It was molar ratio which we have now specified on line 595.

In the methods, the buffer composition for the mass photometry measurement should be indicated.

Good point. We added this on lines 632-633.

Referees cross-commentingI agree to the other reviewers' remarks.Reviewer #2 (Significance (Required)):Chikungunya virus is a very important human pathogen, and research on the architecture of its replication/transcription organelle holds great promise for the development of future therapies. Laurent and colleagues advanced this field by providing pioneering low-resolution 3D structures of the membrane-bound viral protein complex and the viral RNA content of this organelle in situ. In addition, they also assessed the lipid requirements for membrane interaction of the primary viral membrane anchor of this complex, nsP1, in vitro. Underlining the importance of these results, a competing group submitted a partially overlapping study to BioRXiv three months ahead (https://doi.org/10.1101/2022.04.08.487651). Whereas the competing group describes the structure of the neck complex at a much higher resolution, it neither analyzes the RNA content of the spherules nor does it address the lipid preferences of nsP1. The present study by Laurent and colleagues should therefore be of great interest to many virologists and cellular biologists.I am a structural virologist with a focus on envelope glycoproteins. Of relevance to this review, I have experience with cellular electron cryo-tomography and sub-tomogram averaging, as well as in-vitro protein/liposome interaction assays. I do not feel qualified to evaluate the details of the mathematical model for membrane remodeling that is used in the last Results section of this manuscript.

We thank reviewer 2 for this positive overall assessment of our work.

Reviewer #3 (Evidence, reproducibility and clarity (Required)):This is an interesting and well written paper describing the replication spherules generated by Chikungunya virus. Cryo-electron tomography was used to determine a low-resolution structure of the spherule, suggesting that nsP1 is located at the neck of the spherule. Segmentation of the tomograms combined with mathematical modeling was used to produce a structural model for RNA organization in the spherule, suggesting that each spherule contained approximately one copy of a full double-stranded RNA genome. I have a few minor comments:

We are thankful for this positive overall assessment of our work.

The structural studies were complemented with lipid binding assays, showing that nsP1 has an affinity for anionic lipids. While interesting, the connection of these experiments to the rest of the study seems tenuous. There is no further mention of them in the discussion or how they relate to the tomography and their replication model.

We agree that those data were not as well integrated into the paper as they could have been, and are thankful that the reviewer pointed this out. To improve the integration of these data into the manuscript, we have expanded on two ways in which the reconstitution data relate to the rest of the paper: (i) the tomography led us to hypothesise that nsP1 recruits other nsPs to the membrane, which we could confirm with the reconstitution (lines 151-152, and throughout that paragraph), and (ii) the lipid preferences of nsP1 that we could measure using the titrating pulldown experiments inform the possible models for how the spherule membrane is remodeled since nsP1 binds lipids that cannot on their own stabilize a neck shape (lines 217-222). We have also slightly expanded the discussion of the biochemistry and its relation to other data in the paper (lines 307-311).

It is a nice match between the calculated length of the RNA (assumed to be ds) and the length of the vector, but the segmentation of the RNA is not completely convincing based on the provided images. It is difficult to distinguish the RNA strands from the noise and other components in the spherule and, at least by eye, the segments do not seem very connected. Please provide some more details on the tracing algorithm. Has it been validated on a known system?

We appreciate this comment and recognise that we did not sufficiently explain the tracing algorithm. This software was in fact custom written (by others, ca 10 years ago) for cryo-electron tomography and has since been used by others in several studies of cellular cryo-electron tomograms, e.g. to study actin cytoskeleton. We now mention this in the results (lines 195-196) and methods (lines 462-463).

The tomogram video is nice, but it would be good to see a raw image as well, preferably covering a wider view that includes the whole cell, as well as a tomogram that represents the entire field of the reconstruction.

This is a good suggestion. We unfortunately cannot provide images covering the entire cells since this is beyond the field of view of the electron microscope (and an image montage was not acquired at the time of data collection). However, we are now providing an additional supplementary movie that shows the entire field of view of the tomogram. In addition, we have uploaded two of the tomograms (including the uncropped tomogram from Figure 1) to EMDB where they will be downloadable by everyone after publication. We hope the reviewer appreciates that this is all that is technically possible at the moment.

In figure 2, the panels are mislabeled relative to the legend, which refers to the color guide as its own panel.

Thanks for pointing this out, we have rectified this in the revised Figure 2 and its legend.

Line 405: C36 symmetry? Why? Shouldn't it be C12 symmetry?

36-fold symmetry was applied to the lipid membrane part to smoothen it further. The membrane part of the structure is simply outlining the neck shape and this is better visualised in this smoothened representation as also done e.g. in the study of the coronavirus neck complex (Wolff et al., Science 2020). We changed the methods text to make this more clear (line 449).

Line 409: "fit" should be "fitted"

Thanks, Corrected in the revised manuscript line 454.

Referees cross-commentingI think we are all in good agreement, and I believe that the concerns raised can be addressed though a better explanation of the methods and improved discussion of their results.

We also agree and believe we have addressed all of the remaining concerns in the revised manuscript.

Reviewer #3 (Significance (Required)):This is a rather focused study, showing tomography data on the alphavirus replication complex. The main significance of the study is the description of the spherule's dimension and its relationship to the nature of the RNA, which provided a model for the replication process. While somewhat narrow in scope, the study should be of interest to people working in the virus replication and virus structure field. The lipid data are interesting, but does not seem well integrated with the rest of the study.

[Editors' note: further revisions were suggested prior to acceptance, as described below.]

When preparing your revisions, please also address the following points, still outstanding from the reviewers:1. To strengthen the argument that the spherules contain dsRNA, the authors should add discussion of the following points:– It has previously been shown by immunofluorescence imaging that Alphavirus replication sites contain high loads of dsRNA (https://journals.asm.org/doi/full/10.1128/JVI.00085-11).– The viral nsP2 protein, which the authors show is part of the neck complex, has RNA helicase activity and therefore likely plays a role in unwinding the dsRNA genome for replication and transcription (https://www.sciencedirect.com/science/article/pii/S0021925820440335?via%3Dihub).

These are excellent points which we have addressed in the revised manuscript, lines 197-200.

2. The following two interpretations of the same data still seem contradictory.line 208:"Given the limited signal-to-noise ratio of cryo-electron tomograms it seems parsimonious to assume that all spherules in fact contain exactly one full-length copy of the viral genome, of which the tracing algorithm is able to detect 80-90%."lines 257-263:"As noted above, the experimentally measured RNA length likely underestimates the actual RNA length by 10-20%. […] A systematic underestimation of the RNA length would shift the curve slightly, but not change its slope […]"To the best of my understanding, I disagree: if each spherule indeed had the same nucleic acid content, as seemingly implied by the first statement, then the slope of the curve from the second statement should equal zero! This requires clarification.

We agree that our previous manuscript was unclear in this respect, and are thankful that we now got a chance to revise the interpretation. Indeed, if each spherule measured in Figure 4B contained a full copy of the genome in dsRNA form there should – according to our own model – be no difference in the size of mature spherules. The more logical interpretation, which we put forward in the revised text, is that the spherules contain one full copy of the negative (template) strand, most but not all of which is present in dsRNA form. The single-stranded parts would exert less pressure on the membrane by virtue of their smaller volume and much shorter persistence length, which would account for the difference in size with the difference in amount of dsRNA traced. This is now mentioned on lines 191, 215-216 and 264-268 in the revised manuscript.